# Granger causal inference on DAGs identifies genomic loci regulating transcription

**Rohit Singh**[†,1,*]**, Alexander P. Wu**[†,1] **& Bonnie Berger**[1,2,*]
[1] Computer Science and Artificial Intelligence Laboratory, MIT, Cambridge, MA 02139, USA
[2] Department of Mathematics, MIT, Cambridge, MA 02139, USA
`{rsingh,alexwu,bab}@csail.mit.edu`

[†] Authors contributed equally to this work

## Abstract

When a dynamical system can be modeled as a *sequence* of observations, Granger causality is a powerful approach for detecting predictive interactions between its variables. However, traditional Granger causal inference has limited utility in domains where the dynamics need to be represented as directed acyclic graphs (DAGs) rather than as a linear sequence, such as with cell differentiation trajectories. Here, we present GrID-Net, a framework based on graph neural networks with lagged message passing for Granger causal inference on DAG-structured systems. Our motivating application is the analysis of single-cell multimodal data to identify genomic loci that mediate the regulation of specific genes. To our knowledge, GrID-Net is the first single-cell analysis tool that accounts for the temporal lag between a genomic locus becoming accessible and its downstream effect on a target gene's expression. We applied GrID-Net on multimodal single-cell assays that profile chromatin accessibility (ATAC-seq) and gene expression (RNA-seq) in the same cell and show that it dramatically outperforms existing methods for inferring regulatory locus–gene links, achieving up to 71% greater agreement with independent population genetics-based estimates. By extending Granger causality to DAG-structured dynamical systems, our work unlocks new domains for causal analyses and, more specifically, opens a path towards elucidating gene regulatory interactions relevant to cellular differentiation and complex human diseases at unprecedented scale and resolution.[1]

## 1 Introduction

Understanding the structure of a multivariate dynamical system often boils down to deciphering the causal relationships between its variables. Since inferring true causality is often infeasible, requiring additional experiments or independent mechanistic insight, statistical analysis of observational data to identify predictive relationships between the system's variables can be very valuable. The framework of Granger causality does exactly that: in a dataset where observations are temporally ordered, a time-dependent variable $x$ (with value $x_t$ at time $t$) is said to "Granger cause" a second variable $y$ if the history of $x$ at time $t$ (i.e., $x_1, \ldots, x_{t-1}$) is significantly predictive of $y$ at time $t$ even after accounting for $y$'s own history (Granger, 1969; Shojaie & Fox, 2021). Originating in econometrics, Granger causality has been a powerful tool in many domains including biology, finance, and social sciences (Fujita et al., 2007; Yao et al., 2015; Benhmad, 2012; Rasheed & Tahir, 2012). The prerequisite for applying Granger causality, however, is that there be a clear sequential ordering of observations, i.e., the data must conform to a *total* ordering along time.

Often, only a *partial* ordering of the observations is possible. For instance, cell differentiation trajectories may have branches. In text-mining, the citation graph of publications captures the flow of ideas and knowledge. In such cases, the dynamics of the system are more suitably represented as a directed acyclic graph (DAG) corresponding to the partial ordering, with nodes of the DAG representing dynamical states and its edges indicating the flow of information between states.

---

[*]Co-corresponding authors: `{rsingh,bab}@csail.mit.edu`
[1]The code for GrID-Net is available at `https://github.com/alexw16/gridnet`.

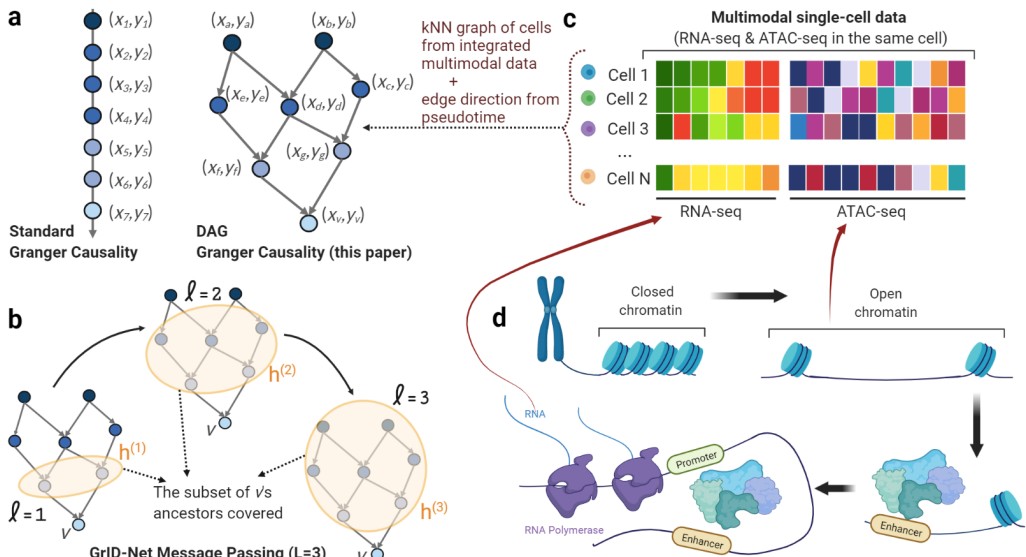

**Figure 1:** OVERVIEW (**a**) We extend the concept of Granger causality, previously applied only to sequentially ordered observations, to partial orderings, i.e., where the information flow can be described as a directed acyclic graph (DAG). (**b**) GrID-Net, our approach for inferring Granger causality on a DAG-structured system, is a graph neural network framework with lagged message-passing ($\ell, L, \boldsymbol{h}$ are defined in Sec. 2.2). (**c**) To identify genomic loci whose accessibility Granger causes the expression of a specific gene, we apply GrID-Net on multimodal single-cell data where chromatin accessibility (ATAC-seq) and gene expression (RNA-seq) have been profiled in the same cell. We model the lag between them by relating the RNA-seq readouts of a cell not to the ATAC-seq readouts of the same cell but to ATAC-seq readouts of cells slightly earlier in the profiled biological process, applying GrID-Net on a DAG constructed from the kNN graph of cells with edges oriented as per pseudotime. (**d**) GrID-Net leverages the biological intuition that the accessibility of a specific genomic locus (ATAC-seq) precedes the binding of regulator proteins to the locus, thus Granger causing the expression levels for its targeted gene (RNA-seq) to be changed.

The key conceptual advance of this work is extending the applicability of Granger causality to systems with partially ordered observations (**Figure** 1a). On a DAG where each node takes on multivariate values, we define a variable $x$ as Granger causing another variable $y$ if the latter's value $y_v$ at the DAG node $v$ can be significantly predicted by the values of $x$ at the ancestor nodes of $v$, even after accounting for $y_v$'s own ancestral history. We introduce GrID-Net (**Gr**anger **I**nference on **D**AGs), a graph neural network (GNN) framework for Granger causal inference in dynamical systems that can be represented as DAGs (**Figure** 1b). We modify the standard GNN architecture to enable lagged message-passing on the DAG, enabling us to accumulate past information for a variable according to the DAG's structure. This, combined with a mathematical formulation of Granger causality more amenable to neural networks, allows GrID-Net to recover nonlinear long-range Granger causal dependencies between the variables.

While our problem formulation is fully general and GrID-Net is broadly applicable, our motivating application is inferring noncoding genomic loci that influence the expression of a specific gene. We apply GrID-Net to single-cell multimodal studies that assay chromatin accessibility (ATAC-seq) and gene expression (RNA-seq) in the same cell, seeking to identify Granger causal relationships between the accessibility of individual chromatin regions (ATAC-seq "peaks") and the expression of specific genes. The system dynamics are represented by a DAG where each node corresponds to a cell. Edges connect cells in similar dynamic states (i.e. similar ATAC-seq and RNA-seq profiles), with edge directions estimated by a pseudotime analysis (Haghverdi et al., 2016) (**Figure** 1c).

Our work addresses a pressing need to identify temporally causal relationships between noncoding loci and gene expression, as traditional approaches for doing so are expensive and slow. The latter include population genetics techniques like expression quantitative trait locus (eQTL) studies, in which human genetic variability at a particular genomic locus is associated with expression changes in the gene of interest. Alternatively, perturbation-based approaches directly seek to identify gene expression changes in response to CRISPR alterations of specific noncoding genomic loci (Fulco et al., 2016). In contrast, we capitalize on the emergence of single-cell multimodal assays that

profile ATAC-seq and RNA-seq simultaneously and offer a new way of discovering such regulatory relationships. Rather than requiring the large population sizes of eQTL studies or being limited to specific noncoding loci as in the perturbation-based approaches, these multimodal experiments enable unbiased high-throughput, genome-wide estimations of peak–gene associations.

However, leveraging this data will require algorithmic innovations. Over $99.5\%$ of ATAC-seq observations ("peaks") in a cell are reported as empty (Singh et al., 2021; Zhang et al., 2019), and this sparsity constrains current approaches. A key limitation of existing methods is also that they are based on correlation estimates and cannot capture the dynamics governing the temporally causal relationship between chromatin accessibility and gene expression. For a noncoding locus to influence a gene's expression, the locus typically must first become accessible, upon which regulator proteins act on it to drive a change in gene expression (**Figure** 1d). This causal mechanism, therefore, entails temporal asynchrony between the ATAC-seq and RNA-seq modalities (Lara-Astiaso et al., 2014; Ostuni et al., 2013). GrID-Net is specifically designed to recover these dynamics between peaks and genes while being robust to the noise and sparsity in single-cell data.

We apply GrID-Net on three single-cell multimodal datasets that characterize a range of dynamic processes, including cancer drug response and cellular differentiation. We show that GrID-Net substantially outperforms current methods in identifying known peak–gene links. To our knowledge, this work presents both the first framework for Granger causal inference on dynamical systems represented as DAGs and the first single-cell method to explicitly model the temporal asynchrony between chromatin accessibility and gene expression.

## 2 METHODS

### 2.1 BACKGROUND ON GRANGER CAUSALITY: STANDARD FORMULATIONS

In time series analyses, the classical Granger causality framework uses the vector autoregressive (VAR) model, which can be expressed as follows (Lütkepohl, 2005).

$$y_t = \sum_{\ell=1}^{L} a_1^{(\ell)} y_{t-\ell} + \sum_{\ell=1}^{L} a_2^{(\ell)} x_{t-\ell} + \epsilon_t \tag{1}$$

Here, $y_t, x_t \in \mathbb{R}$ are values at time $t$ for two stationary time series $y$ and $x$, respectively; $a_1^{(\ell)}$ and $a_2^{(\ell)}$ are coefficients that specify how lag $\ell$ affects the future of time series $y$; and $\epsilon_t$ is a zero-mean noise term. In this model, $y_t$ is assumed to be a linear combination of the $L$ most recent values of $y$ and $x$ each, with $x$ said to Granger cause time series $y$ if and only if $a_2^{(\ell)} \neq 0$ for all $\ell$.

A more generalized formulation — the one we follow — for performing Granger causal inference is to consider two related models for forecasting time series $y$. A *full* model considers the past values of both $y$ and $x$ to forecast $y$, while a *reduced* model excludes the effect of $x$, only containing terms related to the past values of $y$.

$$y_t = f^{(full)}(y_{t-1}, ..., y_{t-L}; x_{t-1}, ..., x_{t-L}) + \epsilon_t \tag{2}$$

$$y_t = f^{(reduced)}(y_{t-1}, ..., y_{t-L}) + \epsilon_t \tag{3}$$

Here, $f^{(full)}(\cdot)$ and $f^{(reduced)}(\cdot)$ are generalized functions that specify how the value of $y_t$ depends on past values of $y$ and $x$. The predictions of the two models are then compared, upon which a Granger causal relationship is declared if the full model's predictions are significantly more accurate.

### 2.2 GRANGER CAUSALITY ON A DAG: GRAPH NEURAL NETWORK FORMULATION

Let the data be represented by a DAG $\mathcal{G} = (\mathcal{V}, \mathcal{E})$ with $n = |\mathcal{V}|$ nodes (i.e., observations) and directed edges $\mathcal{E}$ indicating the information flow or partial order between these observations. For instance, when applied to the case of standard Granger causal inference on time series data, $\mathcal{G}$ would be the linear graph corresponding to the time series. Let $\boldsymbol{y}, \boldsymbol{x} \in \mathbb{R}^n$ correspond to the values of variables $y$ and $x$ on the nodes in $\mathcal{V}$, with $x$ putatively Granger-causing $y$. To infer Granger causality, we compare the full and reduced models as above:

$$\boldsymbol{y} = g(\tilde{\boldsymbol{h}}_y^{(full)} + c\tilde{\boldsymbol{h}}_x^{(full)}) + \boldsymbol{\epsilon} \tag{4}$$

$$\boldsymbol{y} = g(\tilde{\boldsymbol{h}}_y^{(reduced)}) + \boldsymbol{\epsilon} \tag{5}$$

Here, $\tilde{\boldsymbol{h}}_y^{(full)}, \tilde{\boldsymbol{h}}_y^{(reduced)}, \tilde{\boldsymbol{h}}_x^{(full)} \in \mathbb{R}^n$ represent the historical information of $y$ or $x$ (as denoted by the subscript) that is aggregated by graph neural network (GNN) layers. For a history $\tilde{\boldsymbol{h}}$, the value $\tilde{\boldsymbol{h}}[v]$ at node $v \in \mathcal{V}$ is the information accumulated from $v$'s ancestors in $\mathcal{G}$. While $\tilde{\boldsymbol{h}}_y^{(full)}$ and $\tilde{\boldsymbol{h}}_y^{(reduced)}$ are outputs determined from similar architectures, their learned weights and bias terms end up being different since $\tilde{\boldsymbol{h}}_y^{(full)}$ is influenced by its interaction with $\tilde{\boldsymbol{h}}_x$. The coefficient $c$ mediates this interaction and describes the effect of $x$'s history on $y$. In addition, $\boldsymbol{\epsilon} \in \mathbb{R}^n$ is a zero-mean noise term. We also include $g(\cdot)$ as an optional link function that maps its input to the support of the distribution of variable $y$. In our single-cell application, we set $g$ as the exponential function since the target variable ($\boldsymbol{y}$) represents normalized transcript counts, which are non-negative.

For brevity, we only describe how $\tilde{\boldsymbol{h}}_x^{(full)}$ is computed, with $\tilde{\boldsymbol{h}}_y^{(full)}$ and $\tilde{\boldsymbol{h}}_y^{(reduced)}$ being independently computed analogously. Also, since $\boldsymbol{x}$ occurs only in the full model, for notational convenience we drop its superscript *(full)* below, writing just $\tilde{\boldsymbol{h}}_x$. We express $\tilde{\boldsymbol{h}}_x$ as the mean of the outputs of $L$ consecutive GNN layers, denoting the layerwise outputs as $\boldsymbol{h}_x^{(\ell)}$:

$$\tilde{\boldsymbol{h}}_x = \frac{1}{L} \sum_{\ell=1}^{L} \boldsymbol{h}_x^{(\ell)} \tag{6}$$

$$\boldsymbol{h}_x^{(\ell)} = \begin{cases} \sigma(w_x^{(\ell)} \boldsymbol{A}_+^T \boldsymbol{h}_x^{(\ell-1)} + b_x^{(\ell)}) & \text{if } \ell > 1 \\ \sigma(w_x^{(\ell)} \boldsymbol{A}^T \boldsymbol{x} + b_x^{(\ell)}) & \text{if } \ell = 1 \end{cases} \tag{7}$$

Here, $w_x^{(\ell)}, b_x^{(\ell)} \in \mathbb{R}$ are the per-layer weight and bias terms, respectively. $\sigma(\cdot)$ represents the nonlinear activation in each of the GNN layers, chosen here to be the hyperbolic tangent function. $\boldsymbol{A}$ and $\boldsymbol{A}_+ \in \mathbb{R}^{n \times n}$ are matrices defined by the DAG structure. $\boldsymbol{A}$ is the column-normalized adjacency matrix of $\mathcal{G}$, with $A_{ij} = \frac{1}{d_j}$ if edge $(i, j) \in \mathcal{E}$ and 0 otherwise, where $d_j$ is the in-degree of node $j$. We note that a DAG does not have self-loops, so the diagonal terms of $\boldsymbol{A}$ are zero. In $\boldsymbol{A}_+$, we include these diagonal terms, setting them to the same value as others in the column (i.e., $d_j$ is incremented by 1 in $\boldsymbol{A}_+$ to preserve normalization).

Together, $\boldsymbol{A}$ and $\boldsymbol{A}_+$ allow us to extend the key intuition of Granger causality to DAGs: predicting a value at node $v$ from the values of its ancestors that are within $L$ steps. The first layer introduces the lag, using information at the parents of $v$ but not the information at $v$ itself. Each subsequent layer introduces the preceding set of parents, with the diagonal term in $\boldsymbol{A}_+$ ensuring that information already stored at $v$ is integrated with the new information propagated via $v$'s parents. The sequence of GNN layers therefore reflects the successive aggregation of past information by traversing $L$ steps backward along the DAG for each node in the graph. Thus, we are able to conceptually match time series-based formulations (e.g., Eqn. 1) but with the crucial advantage of leveraging the richness of a DAG's structure in directing the information flow and aggregating sparse data.

## 2.3 COMPARING FULL AND REDUCED MODELS TO INFER GRANGER CAUSALITY

Let $\Theta_y^{(full)}, \Theta_y^{(reduced)}, \Theta_x^{(full)} \in \mathbb{R}^{2L}$ denote the set of parameters for the full and reduced models, with $\Theta_x^{(full)} = \{w_x^{(1)}, ..., w_x^{(L)}; b_x^{(1)}, ..., b_x^{(L)}\}$ and $\Theta_y^{(full)}, \Theta_y^{(reduced)}$ defined analogously. All the $\Theta$ parameters are jointly learned by minimizing the combined loss of the full and reduced models (Montalto et al., 2015):

$$\mathcal{L}^{(total)} = \sum_{x,y \in \mathcal{P}} \left( \sum_{v \in \mathcal{V}} \mathcal{L}(\hat{y}_v^{(full)}, y_v) + \sum_{v \in \mathcal{V}} \mathcal{L}(\hat{y}_v^{(reduced)}, y_v) \right) \tag{8}$$

where $\hat{y}_v^{(full)}$ and $\hat{y}_v^{(reduced)}$ correspond to the predictions for observation $v$ by the full and reduced models, respectively. We note that the full and reduced models have completely separate parameters

(see discussion in Appendix A.8). Also, in a multivariate system, not all pairwise combinations of variables may be relevant: $\mathcal{P}$ is the subset of $x, y$ variable pairs whose putative Granger-causal interactions are of interest.

To infer Granger causal interactions between $y$ and $x$, we compare the set of loss terms associated with the full model $\mathcal{L}(\hat{y}_v^{(full)}, y_v)$ to those of the reduced model $\mathcal{L}(\hat{y}_v^{(reduced)}, y_v)$, noting that the precise functional form of $\mathcal{L}$ depends on the domain. If $x$ does not Granger cause $y$, the full and reduced models will have similar explanatory power and hence similar loss values. Otherwise, the full model will have a lower loss. We assess this by a one-tailed F-test, comparing the residual sum of squares of the reduced and full models; an alternative assessment using Welch's $t$-test yielded very similar results (Appendix A.7). We rank the set of candidate $x$–$y$ interactions by the F-statistic, with a higher score corresponding to stronger evidence for a Granger causal interaction.

### 2.4 DOMAIN-SPECIFIC MODEL CUSTOMIZATION

The loss $\mathcal{L}$ should be chosen as appropriate for the domain. In our single-cell context, $y$ corresponds to gene expression. We preprocessed RNA-seq transcript counts, normalizing and log transforming them so that the mean squared error loss was appropriate; we note that most methods based on Pearson correlation of gene expression also seek to minimize squared loss, explicitly or implicitly.

The GrID-Net model can also be customized to account for different lags and lookbacks. The number of GNN layers $L$ corresponds to the maximum amount of past information desired. Similarly, to introduce a $k$-hop lag, the first $k$ GNN layers would use the matrix $\boldsymbol{A}$ while the later layers would use $\boldsymbol{A}_+$. In the sections above, we described a one-hop lag that we have used for all analyses here.

### 2.5 TRAINING DETAILS AND HYPERPARAMETERS

GrID-Net models were trained using the Adam optimizer with a learning rate of 0.001 for 20 epochs or until convergence (defined to be the point at which the relative change in the loss function is less than $0.1/|\mathcal{P}|$ across consecutive epochs). A minibatch size of 1024 candidate peak–gene pairs was used during training, and trainable parameters in the model were initialized using Glorot initialization (Bengio & Glorot, 2010). All GrID-Net models consisted of $L = 10$ GNN layers; the architectures of the three sub-models ($\tilde{\boldsymbol{h}}_y^{(reduced)}$, $\tilde{\boldsymbol{h}}_y^{(full)}$, and $\tilde{\boldsymbol{h}}_x^{(full)}$) were identical but separate. All models were implemented in PyTorch and trained on a single NVIDIA Tesla V100 GPU.

## 3 RELATED WORK

**Single-cell multimodal methods:** The emergence of single-cell multimodal assays has led to the development of tools to leverage the joint profiling of multiple genomic, epigenetic, and functional features in each cell. Many of these tools aim to synthesize the multiple modalities within a dataset to perform downstream analyses that elucidate cell state or gene programs (Singh et al., 2021; Argelaguet et al., 2020; Gayoso et al., 2021). Meanwhile, efforts to detect associations at the level of specific features across modalities have thus far been limited. Current approaches for inferring peak–gene associations predominately rely on simply calculating correlations between these two modalities (Ma et al., 2020; Zhu et al., 2019; Liu et al., 2019). Not only do these correlation-based approaches fail to account for the temporal asynchrony between chromatin accessibility and gene expression, but they are also sensitive to the inherent noisiness of single-cell data (Zhang et al., 2019). These limitations point to the need for going beyond analyzing mere correlations to leveraging the joint dynamics of these features in causal inference frameworks, like Granger causality.

**Granger causality:** Recent approaches for extending Granger causality have primarily focused on enabling the detection of nonlinear or graph-based interactions in multivariate time series data. One category of methods involves nonlinear kernel-based regression models (Marinazzo et al., 2008; Ren et al., 2020); another seeks to learn graphs that describe causal relationships between variables (Basu et al., 2015; Shojaie & Michailidis, 2010). Neural network-based Granger causality models have also more recently been proposed to account for more generalized nonlinear interactions (Tank et al., 2018; Marcinkevičs & Vogt, 2021). Of these methods, efforts have been made to leverage specific neural network architectures to more fully take advantage of the sequential ordering of observations in time series data (Khanna & Tan, 2020; Nauta et al., 2019). Customized Granger causal approaches have also been designed for specific biological applications (Finkle et al., 2018).

As a clarification, we distinguish the DAGs referred to in this paper from the concept of "causal graphs". Here, each node in the DAG corresponds to a state with an associated multivariate observation, and edges describe the ordering between states. In contrast, a causal graph describes a data generation process with nodes corresponding to the dynamical system's variables. In our single-cell peak–gene inference application, performing Granger causal inference on a DAG allows us to capture the diversity of cell states as well as their dynamics. We note that using the standard Granger causality framework would require forcing the rich heterogeneity of a single-cell dataset into a sequential ordering of cells along pseudotime, a suboptimal choice due to the possibility of multiple differentiation trajectories, substantial biological and technical variation in single-cell data (Zhang et al., 2019), and multiple cells representing very similar cell states (Baran et al., 2019).

We also note that a limitation of most Granger causal analyses — including ours — is the inability to rule out hidden confounders that might mediate the interaction between $x$ and $y$ (Mastakouri et al., 2021). Nonetheless, such analysis remains valuable in our peak–gene inference task, as it enables the identification of peak–gene relationships that are temporally causal, upon which a shortlist of hypotheses can be generated for subsequent perturbational or population genetics studies. More importantly, even indirect peak–gene interactions are biologically meaningful. In future work, other approaches for causal inference (e.g., Entner & Hoyer (2010); Pfister et al. (2019)) including Bayesian inference (Brodersen et al., 2015; Glymour et al., 2019) or structural equation models (Spirtes & Zhang, 2016) could be applied to address this limitation.

## 4 RESULTS

**Datasets and Preprocessing**  We analyzed three single-cell multimodal datasets with per-cell ATAC-seq and RNA-seq profiles that characterize a range of dynamical systems, including cell differentiation and drug-treatment responses (Cao et al., 2018; Chen et al., 2019; Ma et al., 2020). The **sci-CAR** dataset profiled 3,260 human lung adenocarcinoma-derived A549 cells after dexamethasone treatment. The **SNARE-seq** dataset evaluated 5,081 cells from the mouse neonatal cerebral cortex. Lastly, the **SHARE-seq** dataset contained 34,774 cells from differentiating skin cells in mouse. We applied geometric sketching (Hie et al., 2019) to identify a representative subset of 5,000 cells for the SHARE-seq dataset, which we use for all downstream analyses. These datasets are high-dimensional (about $200,000$ peaks and $30,000$ genes per cell) and sparse (over $99.5\%$ of peaks in a cell have zero counts). For each dataset, GrID-Net produces results that are specific to the cell type/state composition of the study. While we chose these studies to be tissue-specific, there remains some diversity in cell type and state due to cell-differentiation and perturbation-response variability. To flexibly hone in on specific cell types within a study, users can adapt GrID-Net by simply filtering the dataset to preserve only the cells of interest.

Following recommended practice (Hie et al., 2020), we preprocessed each dataset by applying $\log(1+\text{CPM}/10)$ and $\log(1+\text{CPM}/100)$ transformations to the raw RNA-seq and ATAC-seq count data, respectively; here CPM indicates counts per million. The transformed RNA-seq data for each gene was also divided by the maximum value for that gene. For each single-cell multimodal dataset, we then compiled a set of candidate peak–gene pairs (i.e., $\mathcal{P}$ in Eqn. 8) by selecting ATAC-seq peaks within 1 Mb from each gene (Appendix A.3). We chose this genomic-distance cutoff because almost all enhancer-gene pairs reported in a recent benchmark study were within 1 Mb (Moore et al., 2019). After this filtering, we evaluated 507,408, 916,939, and 872,039 candidate peak–gene pairs for the sci-CAR, SHARE-seq, and SNARE-seq datasets, respectively. Training GrID-Net on these sets of candidate peak–gene pairs took roughly 2–6 hours and required a maximum of 40 GB of RAM. We further discuss scalability, runtime and memory usage details along with recommendations for using geometric sketching (Hie et al., 2019) for large datasets in Appendix A.12.

**DAG construction**  For each of the three multimodal datasets, a kNN graph was constructed on cell representations that unified information from both gene expression and chromatin accessibility via Schema (Appendix A.1, Singh et al. (2021)); other approaches for integrating data modalities (e.g., manifold alignment (Cao et al., 2020)) could also be used here. We next inferred a pseudotime value for each cell. Edges in the kNN graph were then retained only if they aligned with the direction of increasing pseudotime, thus ensuring that no cycles exist and the resulting graph is a DAG. In our results below, we used $k$=15 to build the kNN graph and the diffusion pseudotime algorithm (Haghverdi et al., 2016) to infer pseudotime stamps, although we found GrID-Net to be robust (Ap-

pendix A.2) to the choice of $k$ and other approaches for inferring pseudotime, specifically, Palantir (Setty et al., 2019) and Monocle (Qiu et al., 2017).

**Evaluating causality by using independent information**  A key contribution of this work is a framework for systematically evaluating putative Granger causal peak–gene relationships; to our knowledge, no such framework currently exists. Since ground truth genome-scale perturbational data is unavailable, we propose that the prediction of peak–gene associations be compared against independent, cell type-matched information sources. One information source we use are eQTLs from cell types matched to those profiled in the single-cell studies. An eQTL is a locus-gene pair that associates naturally occurring genetic variability at a locus to gene expression variations observed in large populations, thus serving as a proxy for direct perturbation of the locus (Nica & Dermitzakis, 2013). We also evaluate against cell type-matched chromatin interaction (Hi-C/5C) data that report pairs of genomic loci that are spatially close in the 3D structure of the genome (Schoenfelder & Fraser, 2019). Intuitively, chromatin interaction and accessibility are both manifestations of the underlying gene regulatory process so that, compared to random peak–gene pairs, causal pairs should more likely be in close spatial (i.e., Hi-C) proximity.

We note that, in absolute terms, a high overlap of Granger causal peak–gene pairs with these datasets should not be expected. Like ATAC-seq, both eQTLs and chromatin interaction capture the underlying biology only indirectly and are themselves prone to errors: eQTLs are confounded by correlated mutations elsewhere in the genome and Hi-C/5C data suffers from similar sparsity issues as ATAC-seq. Therefore, we focus here on a relative evaluation, comparing GrID-Net's overlap with these datasets against that of alternative approaches. Also, while good overlap with both eQTL and chromatin interaction datasets is desirable, we believe emphasis should be given to eQTLs since they are the better proxy for an actual perturbation.

**Benchmarking**  We evaluated two correlation-based techniques used in previous single-cell multimodal studies (Cao et al., 2018; Chen et al., 2019; Ma et al., 2020): a) Pearson correlation (**Pearson Corr**) of peak counts and gene expression, with each cell considered an independent observation; and b) Pearson correlation computed after averaging the RNA-seq and ATAC-seq profiles of each cell and its 50 neighbors in the kNN graph of cells (**Pseudocell Corr**). The pseudocell approach, currently the state-of-the-art, seeks robustness by replacing point estimates in the ATAC-seq (or RNA-seq) space with local averages. We used the absolute value of the correlation scores to rank peak–gene pairs in our analyses below. To assess the advantage of operating on a DAG for partially ordered data, we also computed a traditional, linear Granger causal estimate using vector autoregression (**VAR Granger**) and a state-of-the-art generalized vector autoregression method for nonlinear Granger causal inference (**GVAR**). For both these methods, data was forced into a total ordering along pseudotime by partitioning the pseudotime range into 100 bins of equal size and averaging the ATAC-seq and RNA-seq profiles across cells assigned to each bin (Appendix A.4). We note that the **Pseudocell Corr** method uses information from just the kNN graph structure, while both the **VAR Granger** and **GVAR** methods use only pseudotime information. By making use of both the kNN graph and pseudotime information, GrID-Net is able to combine their strengths.

**Predicting eQTLs**  We obtained eQTL data from the GTEx project for the human cell types that most closely correspond to the cells from the single-cell multimodal assays (GTEx Consortium, 2013). For the lung adenocarcinoma-derived A549 cells in the sci-CAR dataset, we used eQTL data for human lungs. The SHARE-seq and SNARE-seq studies profiled mouse tissues, for which matching eQTL data is limited. Since human and mouse epigenomes show strong conservation (Gjoneska et al., 2015; Xiao et al., 2012), we used eQTL data for human skin and the cerebral cortex, respectively (see Appendix A.5), and mapped this data to the mouse genome. For each dataset, we retained peak–gene pairs with a matching locus–gene pair in the eQTL data.

To assess the effectiveness of GrID-Net in predicting eQTLs, we labeled peak–gene pairs associated with eQTLs as *true* (eQTL $p < 10^{-10}$) or *false* (eQTL $p > 0.9$), discarding pairs that were not in either category. GrID-Net and the alternative methods were evaluated on their accuracy in predicting the *true* eQTLs based on the ranking of association scores generated by each method for each peak–gene pair. Across all datasets, GrID-Net outperformed the other methods in predicting eQTLs, achieving the highest AUPRC (area under precision-recall curve) and AUROC (area under receiver operating characteristic) in all cases, indicating that peak–gene pairs prioritized by GrID-Net more closely align with evidence from population genetics (Table 1). The non-linear Granger approach

(GVAR) performs slightly better than linear approach (VAR Granger); however, both methods performed substantially worse than GrID-Net, pointing to the advantage of the DAG-based inference approach for such systems over standard Granger-causal inference on forcibly-ordered data.

**Table 1:** eQTL prediction accuracy on three multimodal datasets

| Method | sci-CAR | | SHARE-seq | | SNARE-seq | |
|---|---|---|---|---|---|---|
| | AUPRC | AUROC | AUPRC | AUROC | AUPRC | AUROC |
| Pearson Corr | 0.0594 | 0.514 | 0.0306 | 0.515 | 0.0096 | 0.619 |
| Pseudocell Corr | 0.0591 | 0.512 | 0.0385 | 0.541 | 0.0084 | 0.573 |
| VAR Granger | 0.0542 | 0.493 | 0.0321 | 0.506 | 0.0037 | 0.439 |
| GVAR (nonlinear) | $0.0556 \pm 0.0016$ | $0.497 \pm 0.016$ | $0.0330 \pm 0.0039$ | $0.536 \pm 0.049$ | $0.0040 \pm 0.0004$ | $0.485 \pm 0.070$ |
| **GrID-Net** | $\mathbf{0.0645 \pm 0.0004}$ | $\mathbf{0.530 \pm 0.002}$ | $\mathbf{0.0604 \pm 0.0041}$ | $\mathbf{0.606 \pm 0.019}$ | $\mathbf{0.0198 \pm 0.0088}$ | $\mathbf{0.681 \pm 0.006}$ |

**Predicting chromatin interactions**   We obtained chromatin interaction data for the specific cell types and experimental settings profiled in the three single-cell multimodal datasets (Davis et al., 2018; Dixon et al., 2012; Poterlowicz et al., 2017). Chromatin interactions were determined for a peak–gene pair by identifying the pairs of genomic windows in the corresponding chromatin interaction dataset that overlapped the peak and the transcription start site (TSS) of the gene. We then identified the peak–gene pairs with significant chromatin interactions (Appendix A.6) and compared the accuracy of GrID-Net and alternative methods in predicting these pairs. We found that GrID-Net achieved the highest AUPRC and AUROC values in this prediction task across all datasets, demonstrating its ability to better identify these key functional features in gene regulation (Table 2).

**Table 2:** Chromatin interaction prediction accuracy on three multimodal datasets

| Method | sci-CAR | | SHARE-seq | | SNARE-seq | |
|---|---|---|---|---|---|---|
| | AUPRC | AUROC | AUPRC | AUROC | AUPRC | AUROC |
| Pearson Corr | 0.0105 | 0.502 | 0.0093 | 0.356 | 0.0106 | 0.513 |
| Pseudocell Corr | 0.0103 | 0.496 | 0.0113 | 0.506 | 0.0108 | 0.517 |
| VAR Granger | 0.0101 | 0.491 | 0.0207 | 0.474 | 0.0102 | 0.504 |
| GVAR (nonlinear) | $0.0103 \pm 0.0001$ | $0.509 \pm 0.005$ | $0.0108 \pm 0.0014$ | $0.440 \pm 0.124$ | $0.0104 \pm 0.0003$ | $0.516 \pm 0.013$ |
| **GrID-Net** | $\mathbf{0.0113 \pm 0.0000}$ | $\mathbf{0.550 \pm 0.001}$ | $\mathbf{0.0402 \pm 0.0023}$ | $\mathbf{0.753 \pm 0.010}$ | $\mathbf{0.0115 \pm 0.0002}$ | $\mathbf{0.540 \pm 0.006}$ |

**GrID-Net is more robust to sparse data than existing approaches**   We hypothesized that GrID-Net's message passing procedure, which aggregates sparse and noisy measurements across cells using the DAG's structure, would offer it robustness to the sparsity of single-cell multimodal data. Because of differing true-positive rates between the full set of candidate peak–gene pairs and the sparse data subset (Appendix A.9, Table 15), the relative performance of methods are compared within rather than across settings. We focused on the subset of genes or peaks for which data is especially sparse (fewer than 5% or 2% of cells for genes or peaks, respectively). Evaluating the various methods on the eQTL prediction task on this subset, we found that GrID-Net's outperformance compared to the alternatives was accentuated (Table 3, additional details in Appendix A.9). GrID-Net also noticeably outperforms the other methods in predicting significant chromatin interactions associated with peak–gene pairs consisting of sparse genes or peaks. These results provide further evidence of GrID-Net's utility in extracting insights from sparse single-cell data.

**Table 3:** eQTL and chromatin interaction prediction accuracy for sparse genes and peak (AUPRC)

| Method | eQTL | | | | | | Chromatin interactions | | | | | |
|---|---|---|---|---|---|---|---|---|---|---|---|---|
| | sci-CAR | | SHARE-seq | | SNARE-seq | | sci-CAR | | SHARE-seq | | SNARE-seq | |
| | Sparse Genes | Sparse Peaks | Sparse Genes | Sparse Peaks | Sparse Genes | Sparse Peaks | Sparse Genes | Sparse Peaks | Sparse Genes | Sparse Peaks | Sparse Genes | Sparse Peaks |
| Pearson Corr | 0.0607 | 0.0566 | 0.0472 | 0.0195 | 0.0136 | 0.0078 | 0.0116 | 0.0106 | 0.0140 | 0.0184 | 0.0102 | 0.0088 |
| Pseudocell Corr | 0.0589 | 0.0562 | 0.0567 | 0.0167 | 0.0122 | 0.0058 | 0.0114 | 0.0104 | 0.0116 | 0.0154 | 0.0101 | 0.0089 |
| VAR Granger | 0.0494 | 0.0520 | 0.0445 | 0.0168 | 0.0045 | 0.0029 | 0.0115 | 0.0101 | 0.0049 | **0.0664** | 0.0097 | 0.0087 |
| GVAR (nonlinear) | 0.0512 | 0.0533 | 0.0496 | 0.0261 | 0.0047 | 0.0027 | 0.0113 | 0.0104 | 0.0068 | 0.0117 | 0.0098 | 0.0091 |
| **GrID-Net** | **0.0668** | **0.0628** | **0.0818** | **0.1450** | **0.0218** | **0.0169** | **0.0121** | **0.0113** | **0.0455** | 0.0596 | **0.0116** | **0.0098** |

**Peak–gene pairs prioritized by GrID-Net are supported by TF motif and ChIP-seq data**   To assess the functional relevance of the detected peak–gene pairs, we sought to relate high-scoring peaks to the transcription factors (TFs) that might bind there. We selected the highest scoring 1% of peak–gene pairs from each of GrID-Net and its alternatives. We next applied Homer (Heinz et al., 2010) to identify TF binding motifs that are co-enriched in sets of peaks linked to a particular gene.

The presence of these co-enriched motifs is potentially indicative of coordinated regulatory control, whereby a gene is modulated by multiple TF binding events featuring the same or related sets of TFs (Pott & Lieb, 2015; Carleton et al., 2017). Consequently, the co-occurrence of these motifs serves as a proxy for the functional significance of the proposed peak–gene pairs. Across all datasets, we observed that the peak–gene pairs detected by GrID-Net were associated with 10–50 times more enriched TF binding motifs than the other methods (Table 4), with GrID-Net's outperformance being statistically significant ($p < 10^{-10}$, one-sided binomial test with Bonferroni correction, details in Appendix A.10). These results suggest that GrID-Net may serve a valuable role in unveiling mechanistic insights into TF-gene regulatory control.

We further evaluated the functional importance of detected peak–gene pairs by comparing peaks in these peak–gene pairs to TF binding sites supported by TF ChIP-seq experimental data (Oki et al., 2018). We first obtained TF ChIP-seq data for the specific cell type(s) represented in each single-cell multimodal dataset. For each of these datasets, we then identified the 0.1% of peaks involved in the highest scoring peak–gene pairs and determined the proportion of such peaks that overlapped with ChIP-seq-derived TF binding sites. We observed that peaks prioritized by GrID-Net more consistently overlapped TF binding sites relative to other methods across all three datasets (Table 4). As TF binding sites are indicative of functional regulatory roles for noncoding loci (Valouev et al., 2008), GrID-Net's effectiveness in prioritizing peaks bound by TFs provides additional evidence for its ability to reveal important gene regulatory roles for specific loci in the noncoding genome.

**Table 4:** Functional relevance of peak–gene pairs. **Left:** Number of putative TF-target gene relations detected from top 1% of peak–gene pairs. **Right:** Proportion of prioritized peaks associated with TF binding sites.

| Method | Number of putative TF-target gene relations | | | Proportion of peaks overlapping TF binding sites | | |
|---|---|---|---|---|---|---|
| | sci-CAR | SHARE-seq | SNARE-seq | sci-CAR | SHARE-seq | SNARE-seq |
| Pearson Corr | 2 | 95 | 0 | 0.630 | 0.521 | 0.232 |
| Pseudocell Corr | 16 | 68 | 23 | 0.565 | 0.573 | 0.261 |
| VAR Granger | 2 | 45 | 56 | 0.652 | 0.479 | 0.268 |
| GVAR | 2.9 ±1.9 | 19.9 ±1.1 | 10.2 ±1.3 | 0.703 ±0.070 | 0.528 ±0.049 | 0.245 ±0.033 |
| **GrID-Net** | **173 ± 71** | **226 ± 31** | **1275 ± 55** | **0.862 ± 0.013** | **0.774 ± 0.012** | **0.573 ± 0.004** |

**Investigating a link between genomic distance and regulatory control** We also explored if larger peak–gene genomic distances correspond to greater temporal lags between peak accessibility and gene expression. By varying $L$ (the number of GNN layers), we allowed for greater pseudotime difference between peaks and genes, finding that architectures with larger $L$ did produce a larger proportion of distal (vs. proximal) peak–gene pairs in their top hits (Appendix A.11; Tables 16, 17).

## 5 DISCUSSION

We extend the applicability of Granger causal analysis to dynamical systems represented as DAGs, introducing GrID-Net to perform such inference. GrID-Net takes advantage of the expressive power of graph neural networks, enabling the detection of long-range nonlinear interactions between variables. We focused on applying it to study multimodal single-cell gene regulatory dynamics, given the prevalence of graph-based representations of cellular landscapes in the field. GrID-Net demonstrated substantial improvements in accuracy relative to current methods in predicting peak–gene links from independent, cell type-matched information sources. Capitalizing on the high resolution of ATAC peaks ($\sim$ 1 kb), regulatory links detected by GrID-Net from single-cell multimodal assays may be used to precisely hone in on functional regions within noncoding loci or to complement existing chromatin interaction data towards this goal (Fulco et al., 2019; Nasser et al., 2021). This increased resolution, combined with the genome-wide scale of single-cell multimodal assays, positions GrID-Net to serve as a vital tool for furthering our understanding of key aspects of gene regulatory dynamics relevant to fundamental biology and disease.

Additionally, the ever growing quantities of network data across numerous non-biological domains, including social media networks and financial transaction networks, also points to the broader applicability of GrID-Net. The ability of GrID-Net to capture long-range, nonlinear interactions in DAGs opens the door to novel analyses on the many existing datasets that are characterized by these graph-based dynamics as well.

## ACKNOWLEDGEMENTS

We thank the anonymous reviewers of ICLR 2022 for their valuable feedback and suggestions. Figure 1 was created using biorender.io. All authors were supported by the NIH grant R35GM141861.

## REPRODUCIBILITY STATEMENT

In order to ensure that the full set of results presented in this paper are reproducible, we have included source code for our method, which can be found at `https://github.com/alexw16/gridnet`. In addition, all analyses described in the paper were performed on publicly available datasets, for which we have provided references throughout the main text. Similarly, the software tools used for biology-specific analyses (e.g., Homer) are publicly available, and we have described the specific settings that were used for this study in the "Appendix" section. We have also detailed pre-processing steps for the various single-cell datasets in the main text and included both pre-processing and analysis procedures related to the eQTL, Hi-C/5C, and TF binding motif data in the "Appendix" section. We also made sure to provide details on the specific training details and hyperparameter choices related to training our neural network model in a dedicated subsection in the "Methods" section entitled "Training Details and Hyperarameters".

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

## A  APPENDIX

### A.1  DAG CONSTRUCTION DETAILS

For the three single-cell multimodal datasets, we applied a standard $\log(1+\text{CPM}/10)$ transformation to the raw RNA-seq counts data and a $\log(1+\text{CPM}/100)$ transformation to the raw ATAC-seq counts data, where CPM indicates counts per million. Genes and accessible regions detected in fewer than 0.1% of cells in each dataset were excluded. For each dataset, the expression of each gene was scaled to unit variance and zero mean, after which the top 2000 most highly variable genes (Satija et al., 2015) were retained. In the original SHARE-seq and SNARE-seq studies, topic modeling-based approaches were used to transform ATAC-seq counts data for downstream trajectory analyses, prompting us to transform the ATAC-seq counts in these datasets using *tf-idf*, a procedure derived from text-based topic modeling that has been applied extensively to analyze single-cell ATAC-seq data (Cusanovich et al., a;b;c).

The normalized RNA-seq and ATAC-seq data were then projected onto their top 100 principal components (excluding components that have Spearman correlation $\rho > 0.9$ with the total counts per cell). Schema (Singh et al., 2021) was then used to generate unified representations for each cell that synthesize information from both the RNA-seq and ATAC-seq representations. For the sci-CAR and SNARE-seq datasets, RNA-seq was used as the primary modality in Schema with ATAC-seq acting as a secondary modality. For the sci-CAR dataset, the time of data collection post-drug treatment was used as a tertiary modality. Meanwhile, ATAC-seq served as the primary modality for the SHARE-seq dataset, as the original SHARE-seq study identified strong cell cycle effects represented in the RNA-seq data for certain subpopulations of cells and consequently used ATAC-seq as the modality for analyzing lineage dynamics for the dataset (Ma et al., 2020). We set the minimum correlation between the primary and the remaining modalities to be 0.9 in Schema. For each dataset, a kNN graph ($k = 15$) based on distances between the unified representations generated by Schema was constructed. Pseudotime stamps were then inferred for each cell based on this kNN graph using diffusion pseudotime (Haghverdi et al., 2016). Finally, edges in the kNN graph that cohere with the direction of increasing pseudotime are retained to generate a DAG.

## A.2 DAG ROBUSTNESS ANALYSIS

In addition to the diffusion pseudotime algorithm, we also evaluated Palantir and Monocle for inferring pseudotime. The performance of GrID-Net was robust to the choice of algorithm (Tables 5, 6,7,8).

**Table 5:** eQTL prediction accuracy by pseudotime algorithm

| Method | sci-CAR | | SHARE-seq | | SNARE-seq | |
|---|---|---|---|---|---|---|
| | AUPRC | AUROC | AUPRC | AUROC | AUPRC | AUROC |
| Diffusion | $0.065 \pm 0.0004$ | $0.530 \pm 0.0019$ | $0.060 \pm 0.0041$ | $0.606 \pm 0.0194$ | $0.020 \pm 0.0088$ | $0.681 \pm 0.0055$ |
| Palantir | $0.064 \pm 0.0004$ | $0.530 \pm 0.0018$ | $0.060 \pm 0.0028$ | $0.611 \pm 0.0083$ | $0.019 \pm 0.0068$ | $0.682 \pm 0.0106$ |
| Monocle | $0.065 \pm 0.0004$ | $0.531 \pm 0.0019$ | $0.061 \pm 0.0057$ | $0.605 \pm 0.0266$ | $0.018 \pm 0.0068$ | $0.675 \pm 0.0099$ |

**Table 6:** Chromatin interaction prediction accuracy by pseudotime algorithm

| Method | sci-CAR | | SHARE-seq | | SNARE-seq | |
|---|---|---|---|---|---|---|
| | AUPRC | AUROC | AUPRC | AUROC | AUPRC | AUROC |
| Diffusion | $0.011 \pm 0.0000$ | $0.550 \pm 0.0008$ | $0.040 \pm 0.0023$ | $0.753 \pm 0.0105$ | $0.011 \pm 0.0002$ | $0.540 \pm 0.0060$ |
| Palantir | $0.011 \pm 0.0000$ | $0.550 \pm 0.0009$ | $0.057 \pm 0.0193$ | $0.783 \pm 0.0373$ | $0.011 \pm 0.0001$ | $0.543 \pm 0.0039$ |
| Monocle | $0.011 \pm 0.0000$ | $0.548 \pm 0.0010$ | $0.057 \pm 0.0209$ | $0.793 \pm 0.0525$ | $0.011 \pm 0.0002$ | $0.544 \pm 0.0052$ |

**Table 7:** eQTL prediction accuracy by $k$ in kNN-graph construction

| Method | sci-CAR | | SHARE-seq | | SNARE-seq | |
|---|---|---|---|---|---|---|
| | AUPRC | AUROC | AUPRC | AUROC | AUPRC | AUROC |
| k=30 | $0.0645 \pm 0.0004$ | $0.5303 \pm 0.0019$ | $0.0630 \pm 0.0051$ | $0.6151 \pm 0.0162$ | $0.0196 \pm 0.0087$ | $0.6788 \pm 0.0101$ |
| k=50 | $0.0645 \pm 0.0004$ | $0.5302 \pm 0.0017$ | $0.0629 \pm 0.0069$ | $0.6159 \pm 0.0160$ | $0.0198 \pm 0.0088$ | $0.6778 \pm 0.0108$ |
| k=15 | $0.0645 \pm 0.0004$ | $0.5299 \pm 0.0019$ | $0.0604 \pm 0.0041$ | $0.6060 \pm 0.0194$ | $0.0198 \pm 0.0088$ | $0.6805 \pm 0.0055$ |

## A.3 SINGLE-CELL DATA PREPROCESSING AND CANDIDATE PEAK–GENE PAIR SELECTION

We applied the same CPM and log transformations as above to the raw RNA-seq and ATAC-seq counts data for all three single-cell multimodal datasets. We then scaled the RNA-seq data for each gene by the maximum value for that gene, so that expression values for all genes in each dataset are in the same [0,1] range. The peak–gene pairs considered for each dataset were determined by selecting genes and peaks detected in greater than 0.1% of cells and then retaining all peak–gene pair combinations in which the peak is within 1 Mb of the gene. For the SHARE-seq dataset, we only considered peaks detected in greater than 1% of cells so as to have a comparable set of candidate peak–gene pairs across the three datsets.

## A.4 BENCHMARKING

For the VAR Granger model, peak–gene pairs were scored on the $-\log_{10}(p)$-value associated with the statistical significance of Granger causality, as calculated using the `grangercausalitytests` function with default settings in the `statsmodels` package (Seabold & Perktold, 2010).

## A.5 MATCHING EQTLS WITH SINGLE-CELL ATAC-SEQ & RNA-SEQ DATA

The procedure to match a peak–gene pair with a eQTL variant–gene pair involved identifying all catalogued variants positioned in the peak of interest and selecting the variant with the most statistically significant association with the gene in the peak–gene pair. To perform this procedure for the SNARE-seq and SHARE-seq datasets, we used liftOver (Gonzalez et al., 2021) with default settings to convert human genomic coordinates represented in the GTEx datasets to mouse coordinates. We chose to use human eQTL data here because it is important to obtain tissue-specific eQTL data. Unfortunately, there is relatively little eQTL data available in mouse and, to our knowledge, when available it is organized around different strains of mice rather than tissues (Doss et al., 2005; Hofstetter et al., 2007). On the other hand, the transfer of eQTL knowledge from human to mouse is well-motivated because the epigenome is largely conserved across mouse and human (Gjoneska et al., 2015; Xiao et al., 2012).

**Table 8:** Chromatin interaction prediction accuracy by $k$ in kNN-graph construction

| Method | sci-CAR | | SHARE-seq | | SNARE-seq | |
|--------|---------|---------|-----------|---------|-----------|---------|
| | AUPRC | AUROC | AUPRC | AUROC | AUPRC | AUROC |
| k=30 | $0.0113 \pm 0.0000$ | $0.5499 \pm 0.0009$ | $0.0362 \pm 0.0036$ | $0.7233 \pm 0.0264$ | $0.0114 \pm 0.0002$ | $0.5408 \pm 0.0053$ |
| k=50 | $0.0113 \pm 0.0000$ | $0.5503 \pm 0.0009$ | $0.0351 \pm 0.0046$ | $0.7152 \pm 0.0272$ | $0.0113 \pm 0.0002$ | $0.5373 \pm 0.0057$ |
| k=15 | $0.0113 \pm 0.0000$ | $0.5497 \pm 0.0008$ | $0.0402 \pm 0.0023$ | $0.7527 \pm 0.0105$ | $0.0115 \pm 0.0002$ | $0.5400 \pm 0.0060$ |

## A.6 Chromatin interaction data preprocessing

For the sci-CAR, SHARE-seq, and SNARE-seq datasets, the corresponding chromatin interaction datasets contained scores indicating the frequency of interactions between genomic windows. We retained peak–gene pairs that were matched with the genomic windows presented in the chromatin interaction datasets and chose the most frequent 1% of interactions as the set to be predicted.

## A.7 Comparison of F-test and Welch's t-test to rank putative peak–gene interactions

For each candidate peak–gene pair, we compare the full and reduced models using an F-test, as is common in Granger causal inference. Specifically, we evaluate the sum-of-squared-errors of the reduced and full models as per an F distribution with $(2L + 1, n - 4L - 1)$ degrees of freedom. Here, $n$ is the number of cells, each of the $L$ GNN layers includes two parameters (the full model has two such architectures while the reduced model has one) and $c$ is an additional parameter in the full model. Another approach would be to compare the full and reduced models' losses with the Welch's $t$-test, where the alternative hypothesis is that the mean of the loss terms for the full model is less than the mean of the loss terms for the reduced model. In both approaches, candidate interactions are then ranked by the test statistic with higher scores corresponding to a stronger likelihood of Granger causal interaction.

We compared the performance of the two tests on the eQTL and chromatin interaction prediction tasks, finding that they yielded very similar results (Tables 9,10). This similarity is likely due to our use of the test statistics to rank peak–gene pairs, upon which the top-ranking pairs are selected. Since the degrees of freedom are identical across all candidate pairs, their ranking depends only on the relative losses of the full and reduced models.

**Table 9:** Comparison of F-test and $t$-test on eQTL prediction accuracy

| Method | sci-CAR | | SHARE-seq | | SNARE-seq | |
|--------|---------|---------|-----------|---------|-----------|---------|
| | AUPRC | AUROC | AUPRC | AUROC | AUPRC | AUROC |
| F-test | $0.0645 \pm 0.0004$ | $0.5300 \pm 0.0020$ | $0.0604 \pm 0.0041$ | $0.6060 \pm 0.0190$ | $0.0198 \pm 0.0088$ | $0.6810 \pm 0.0060$ |
| $t$-test | $0.0697 \pm 0.0006$ | $0.5400 \pm 0.0020$ | $0.0573 \pm 0.0047$ | $0.6100 \pm 0.0180$ | $0.0175 \pm 0.0062$ | $0.6770 \pm 0.0080$ |

**Table 10:** Comparison of F-test and $t$-test on chromatin interaction prediction accuracy

| Method | sci-CAR | | SHARE-seq | | SNARE-seq | |
|--------|---------|---------|-----------|---------|-----------|---------|
| | AUPRC | AUROC | AUPRC | AUROC | AUPRC | AUROC |
| F-test | $0.0113 \pm 0.0000$ | $0.5500 \pm 0.0010$ | $0.0402 \pm 0.0023$ | $0.7530 \pm 0.0100$ | $0.0115 \pm 0.0002$ | $0.5400 \pm 0.0060$ |
| $t$-test | $0.0109 \pm 0.0000$ | $0.5460 \pm 0.0010$ | $0.0375 \pm 0.0066$ | $0.7620 \pm 0.0110$ | $0.0115 \pm 0.0003$ | $0.5370 \pm 0.0070$ |

## A.8 Comparison of combined and separated loss functions for full and reduced models

During the GNN's training, the sum of the losses of the full and reduced models is optimized. Since the two models do not share any parameters, optimizing the sum of their losses corresponds to optimizing each loss separately. We chose this joint-optimization strategy for efficiency and to avoid stochastic variations if the models were trained separately. Nonetheless, to confirm that optimizing the sum of losses does not introduce any unintended dependencies, we also evaluated GrID-Net's performance on the eQTL and chromatin interaction prediction tasks when the models were optimized in completely separate training runs. The results for separate and joint training were essentially identical (Tables 11, 12), suggesting that our joint training works as intended.

**Table 11:** Comparison of combined and separated loss functions on eQTL prediction accuracy

| Method | sci-CAR | | SHARE-seq | | SNARE-seq | |
|---|---|---|---|---|---|---|
| | AUPRC | AUROC | AUPRC | AUROC | AUPRC | AUROC |
| Separated | $0.0645 \pm 0.0004$ | $0.530 \pm 0.002$ | $0.0603 \pm 0.0043$ | $0.605 \pm 0.020$ | $0.0201 \pm 0.0085$ | $0.684 \pm 0.006$ |
| Combined | $0.0645 \pm 0.0004$ | $0.530 \pm 0.002$ | $0.0604 \pm 0.0041$ | $0.606 \pm 0.019$ | $0.0198 \pm 0.0088$ | $0.681 \pm 0.006$ |

**Table 12:** Comparison of combined and separated loss functions on chromatin interaction prediction accuracy

| Method | sci-CAR | | SHARE-seq | | SNARE-seq | |
|---|---|---|---|---|---|---|
| | AUPRC | AUROC | AUPRC | AUROC | AUPRC | AUROC |
| Separated | $0.0113 \pm 0.0000$ | $0.550 \pm 0.001$ | $0.0404 \pm 0.0019$ | $0.752 \pm 0.010$ | $0.0115 \pm 0.0002$ | $0.539 \pm 0.006$ |
| Combined | $0.0113 \pm 0.0000$ | $0.550 \pm 0.001$ | $0.0402 \pm 0.0023$ | $0.753 \pm 0.010$ | $0.0115 \pm 0.0002$ | $0.540 \pm 0.006$ |

## A.9 ROBUSTNESS OF GRID-NET TO SPARSITY

Tables 13 and 14 expand upon the results in Table 3 by also showing standard deviations (across three runs) of GVAR's and GrID-Net's AUPRC results; the other methods are fully deterministic and their results do not change across runs. Table 13 shows the baseline proportions of true positives across the different comparison conditions to provide context for differences in the range of AUPRC values across these conditions.

**Table 13:** eQTL prediction accuracy for sparse genes and peak (AUPRC). Accompanies Table 3.

| Method | sci-CAR | | SHARE-seq | | SNARE-seq | |
|---|---|---|---|---|---|---|
| | Sparse Genes | Sparse Peaks | Sparse Genes | Sparse Peaks | Sparse Genes | Sparse Peaks |
| Pearson Corr | 0.0607 | 0.0566 | 0.0472 | 0.0195 | 0.0136 | 0.0078 |
| Pseudocell Corr | 0.0589 | 0.0562 | 0.0567 | 0.0167 | 0.0122 | 0.0058 |
| VAR Granger | 0.0494 | 0.0520 | 0.0445 | 0.0168 | 0.0045 | 0.0029 |
| GVAR (nonlinear) | $0.0512 \pm 0.0014$ | $0.0533 \pm 0.0015$ | $0.0496 \pm 0.0049$ | $0.0261 \pm 0.0040$ | $0.0047 \pm 0.0004$ | $0.0027 \pm 0.0002$ |
| **GrID-Net** | $0.0668 \pm 0.0008$ | $0.0628 \pm 0.0004$ | $0.0818 \pm 0.0051$ | $0.1450 \pm 0.0310$ | $0.0218 \pm 0.0096$ | $0.0169 \pm 0.0082$ |

## A.10 TF BINDING MOTIF ENRICHMENT ANALYSIS

For each gene that was associated with more than 5 peaks in the highest scoring 1% peak–gene pairs, we applied HOMER with default parameters on the genomic windows defined by the set of peaks to test for enrichment of TF binding motifs (Heinz et al., 2010). We used a Benjamini-Hochberg-corrected p-value cutoff of $0.01$ to call enriched TF binding motifs. Against each baseline, the statistical significance of the difference between GrID-Net and the baseline in the number of enriched motifs was assessed via a one-sided binomial test, with Bonferroni correction for multiple hypothesis testing. All comparisons were significant at the $p < 10^{-10}$ level.

## A.11 GRID-NET SUGGESTS POSSIBLE LINK BETWEEN GENOMIC DISTANCE AND TEMPORAL REGULATORY CONTROL

GrID-Net's explicit consideration of the temporal asynchrony between chromatin accessibility and gene expression enables the genome-wide study of lags in gene regulation for the very first time. We classified peak–gene pairs as *proximal* (genomic distance $< 10$kb) or *distal* ($\geq 100$ kb). We then evaluated different architectures of GrID-Net that vary in the number of GNN layers $L$, hypothesizing that architectures with higher $L$ (i.e., allowing greater pseudotime difference between peaks and genes) may detect more distal peak–gene pairs. Inspecting the top 1% of peak–gene pairs detected by different architectures, we found this to indeed be the case: across all datasets, architectures with higher $L$ detected larger ratios of distal-to-proximal peak–gene pairs (Tables 16, 17), suggesting that distal peak–gene interactions may be marked by longer pseudotime difference between peak and genes. This finding suggests the possibility of a previously unreported link between genomic distance and the temporal lag between a regulatory element's accessibility and its target gene's expression. However, like other Granger causal analyses, our approach also can not account for hidden confounders; this is especially pertinent here since the greater pseudotemporal lag between distal peaks and genes may be mediated by other regulatory mechanisms rather than being a direct influence. One hypothesis is that the complex coordination of regulatory factors required

**Table 14:** Chromatin interaction prediction accuracy for sparse genes and peak (AUPRC). Accompanies Table 3.

| Method | sci-CAR | | SHARE-seq | | SNARE-seq | |
|---|---|---|---|---|---|---|
| | Sparse Genes | Sparse Peaks | Sparse Genes | Sparse Peaks | Sparse Genes | Sparse Peaks |
| Pearson Corr | 0.0116 | 0.0106 | 0.0140 | 0.0184 | 0.0102 | 0.0088 |
| Pseudocell Corr | 0.0114 | 0.0104 | 0.0116 | 0.0154 | 0.0101 | 0.0089 |
| VAR Granger | 0.0115 | 0.0101 | 0.0049 | 0.0664 | 0.0097 | 0.0087 |
| GVAR (nonlinear) | $0.0113 \pm 0.0000$ | $0.0104 \pm 0.0001$ | $0.0068 \pm 0.0010$ | $0.0117 \pm 0.0034$ | $0.0098 \pm 0.0003$ | $0.0091 \pm 0.0003$ |
| **GrID-Net** | $0.0121 \pm 0.0001$ | $0.0113 \pm 0.0000$ | $0.0455 \pm 0.0144$ | $0.0596 \pm 0.0177$ | $0.0116 \pm 0.0002$ | $0.0098 \pm 0.0002$ |

**Table 15:** Baseline proportions of true positives in full, sparse gene, and sparse peak comparison conditions.

| Category | sci-CAR | | SHARE-seq | | SNARE-seq | |
|---|---|---|---|---|---|---|
| | eQTL | Chrom Int | eQTL | Chrom Int | eQTL | Chrom Int |
| Full | 0.0558 | 0.0101 | 0.0302 | 0.0111 | 0.00401 | 0.0100 |
| Sparse Genes | 0.0515 | 0.0113 | 0.0452 | 0.0071 | 0.00488 | 0.0095 |
| Sparse Peaks | 0.0535 | 0.0102 | 0.0192 | 0.0111 | 0.00284 | 0.0085 |

to enable distal regulatory interactions is associated with greater lags between distal regulatory elements' accessibility and the gene expression changes they affect (Panigrahi & O'Malley, 2021).

**Table 16:** Percentages of peak–gene pairs within 10kb

| | sci-CAR | SHARE-seq | SNARE-seq |
|---|---|---|---|
| 2 GNN Layers | 14.5% | 6.04% | 2.53% |
| 5 GNN Layers | 14.1% | 5.20% | 2.35% |
| 10 GNN Layers | 13.7% | 5.33% | 2.25% |

## A.12 RUNTIME AND MEMORY USAGE

GrID-Net's runtime and memory usage are dependent on a variety of factors, most notably the number of nodes in the DAG describing the system of interest. In our model, the DAG is represented by the $N \times N$ adjacency matrices $\boldsymbol{A}$ and $\boldsymbol{A}_+$, where $n$ is the number of nodes in the DAG (i.e. number of cells in a multimodal single-cell dataset). These matrices serve as the primary bottlenecks for both runtime and memory usage. Memory usage scales quadratically with $n$, while calculations involving $\boldsymbol{A}$ or $\boldsymbol{A}_+$ can be parallelized on a GPU to reduce the runtime. For very large single-cell datasets, we recommend the use of sketching techniques (Hie et al., 2019; DeMeo & Berger, 2020) so that the matrices will fit into GPU memory. In addition, training time scales linearly with the number of candidate peak-gene pairs. In our results above, we evaluated 507,408, 916,939, and 872,039 candidate peak–gene pairs for the sci-CAR, SHARE-seq, and SNARE-seq datasets, respectively. For these same datasets, we considered systems of 3,260, 5,000, and 5,081 cells. The average total runtime for GrID-Net was 1.8 hours, 5.5 hours, and 5.3 hours, while peak memory usage was approximately 31 GB, 35 GB, and 39 GB for these respective datasets.

**Table 17:** Ratio of distal ($> 100$kb) to proximal ($< 10$kb) peak–gene pairs in the top 1% hits

|  | sci-CAR | SHARE-seq | SNARE-seq |
|---|---|---|---|
| 2 GNN Layers | $6.10 \pm 0.062$ | $12.53 \pm 0.128$ | $35.97 \pm 1.322$ |
| 5 GNN Layers | $6.23 \pm 0.119$ | $15.33 \pm 0.522$ | $37.45 \pm 1.774$ |
| 10 GNN Layers | $6.05 \pm 0.055$ | $15.77 \pm 0.150$ | $41.16 \pm 1.228$ |

