# OpenReview forum: "Granger causal inference on DAGs identifies genomic loci regulating transcription"
_ICLR.cc/2022/Conference — ICLR 2022 Poster_

### Official Review · Reviewer_A4M6 · 2021-10-29

**Correctness:** 4
**Technical Novelty And Significance:** 3
**Empirical Novelty And Significance:** 3
**Recommendation:** 8
**Confidence:** 4

**Main Review:**


One thing that i would like the authors to mention in their paper is that the presented work assumes causal sufficiency; i.e. that there are no hidden confounders in the data. This is a very strict assumption which is also often violated in real datasets. It is important to point out here that any Granger related method suffers from this issue, and for that reason the interpretation of the findings as 'causal' should be careful, as such methods are not able to differentiate between X \rightarrow Y, and X \leftarrow H \rightarrow Y, where H is a potential hidden node acting as a confounder.

I would like to ask some questions the authors:

1. What is the intuition of the authors about the robustness of their method in sparse settings? I see the data driven conclusion,
but i would be interested in the theoretical explanation or some intuition about it from the authors.

2. I appreciate a lot the presentation of the related work and the comparison experiments. Nevertheless, I would like to also see comparison with a non-linear implementation of Granger, as it is not straightforward to understand here whether the poorer performance of VAR Granger comes from the linearity of the method, or from the fact that Granger itselelf does not take into account partial ordering of the data.

3. What is the variance of the results presented in Table 1? Is the difference between GrID-Net and the other methods significant?

4. In the paragraph about methods related to Granger causality (p.5), the authors make an effort to list the related work, however they miss some important causal discovery methods in time series (and lagged panel data in general) methods that outperform Granger and don't require causal sufficiency.
i.e.
 - Mastakouri et al.2020 Necessary and Sufficient Conditions for Causal feature selection in time series with latent common causes.
- D. Entner, and P. O. Hoyer (2010). On causal discovery from time series data using FCI (PGM2010)
- N. Pfister, P. Bühlmann, J. Peters: Invariant Causal Prediction for Sequential Data. Journal of the American Statistical Association, 114(527):1264-1276.

**Summary Of The Paper:**

The paper proposes a modification of Granger causality so that it is applicable in partially ordered observations. Authors achieve that by adapting the architecture of GNNs to enable lagged-passed information in the DAG. Finally they apply it on a real data problem with the goal to infer non-coding genomic loci that causally influence the expression of a specific gene. In comparison with other three methods, the proposed method GrID net outperforms them on an open dataset.


**Summary Of The Review:**

Overall I find the paper to be very well written, with a new method clearly motivated from a real-world problem the field of genomics. For that reason i am inclined for acceptance of the paper. I would like the authors to mention the strict assumption of causal sufficiency that their work assumes and make it clear that their results may suffer from it. I would also like to ask the authors to answer my questions above.

--------------------- update ---------------------------

Since the authors did not respond to any of my comments and concerns above, I decided to reduce my score. Not clearly stating that causal sufficiency is assumed is a major point, as it leads to wrong "causal conclusions". Moreover, the lack of explanation from the authors regarding their intuition in some key points of the paper contributes to lack of clarity. In addition to the above, the lack of comparison with linear Granger or at least discussion about it, make me decrease my score significantly. It is a pity because this could have been a good paper, if the authors have taken the time to answer and fix these points by answering to the reviewers. Without their responses/edits the paper contains strong overstatements.


------------------- update after authors posted their rebuttal -------------------
I am reverting my score to its original value, given that the authors answered all my comments and updated their paper accordingly.

---

> ### Author Response · Authors · 2021-11-23
> **Detailed Responses to Reviewer #4 (A4M6)**
>
> We thank you for your feedback and below are detailed responses to your comments. Please also see a summary of the major revisions above.
>
> **Comment:** One thing that i would like the authors to mention in their paper is that the presented work assumes causal sufficiency; i.e. that there are no hidden confounders in the data. This is a very strict assumption which is also often violated in real datasets.
>
> **Response:** We are grateful for the reviewer’s insight into the effects of hidden confounders in causal inference and have now added a discussion of the limitations of causal analyses imposed by these hidden confounders (“Granger causality” subsection). Despite this limitation, we believe Granger causal approaches like GrID-Net still provide very substantial value. First, GrID-Net helps by generating a shortlist of hypotheses (i.e. putative peak-gene relationships) for subsequent validation by perturbational studies. Second, even if the peak-gene interaction is indirect and mediated by a hidden confounder, disruption of the genetic locus will indirectly result in gene expression changes; this is biologically meaningful.
>
> **Comment:** What is the intuition of the authors about the robustness of their method in sparse settings? I see the data driven conclusion, but i would be interested in the theoretical explanation or some intuition about it from the authors.
>
> **Response:** We appreciate the reviewer’s interest in the intuition behind GrID-Net’s effectiveness in sparse settings and note that direct comparisons of a method’s performance on the full candidate peak-gene set vs. the sparse set are complicated by varying true positive rates between the full set and the sparse subset (see Appendix A.9). As such we believe it is more appropriate to compare relative performance across methods in each setting, rather than absolute AUPRC scores across settings. In this regard, GrID-Net’s improvement over other methods is stronger when evaluated on sparse data. We believe this may be because GrID-Net's use of node ancestors in a DAG essentially has an averaging effect that reduces the effect of noise. Furthermore, if some parts of the DAG are accompanied by sparse data while others are not, GrID-Net is able to take advantage of the regions of the DAG where data is less sparse. We have included an explanation of this intuition in the main text (“GrID-Net is more robust to sparse data than existing approaches”) as well the Appendix A.9.
>
> **Comment:** Comparison with nonlinear Granger causal inference methods
>
> **Response:** We thank the reviewer for recognizing the value of evaluating GrID-Net against nonlinear Granger causality methods and have now included results for GVAR, a state-of-the-art nonlinear Granger causal inference approach. Across our evaluations, we observed that GVAR tended to often outperform the standard linear implementation of Granger causality, suggesting the importance of accounting for nonlinearity in Granger causality inference. At the same time, GrID-Net consistently outperformed GVAR across all metrics, providing evidence of the importance of GriD-Net’s additional usage of temporal information encoded in DAGs.
>
> **Comment:** What is the variance of the results presented in Table 1? Is the difference between GrID-Net and the other methods significant?
>
> **Response:** We have included the means and standard deviations of multiple runs of GrID-Net and GVAR across different parameter initializations. The other baselines were completely deterministic and showed no variation across multiple runs. In most cases of Table 1, GrID-Net’s outperformance is at least 2 standard deviations better than others. Separately, we scored GrID-Net’s performance against baselines on the TF motif discovery study (Table 3) via a binomial test. For each baseline, we considered its motif discovery rate as the null hypothesis, evaluating via a binomial test if GrID-Net’s motif discovery rate was significantly higher. After Bonferroni correction for multiple hypothesis testing, GrID-Net’s outperformance was significant in all cases (p < 10^-10; “Results”).
>
> **Comment:** In the paragraph about methods related to Granger causality (p.5), the authors make an effort to list the related work, however they miss some important causal discovery methods in time series (and lagged panel data in general) methods that outperform Granger and don't require causal sufficiency.
>
> **Response:** We are appreciative of the reviewers bringing to our attention the significance of contextualizing our work in relation to the provided list of references. In the “Granger causality” subsection, we have now added a discussion of the important topics described in these references.

---

> ### Author Response · Authors · 2021-11-26
> **Thank you for updating the score!**
>
> Your feedback was extremely valuable and helped improve the paper substantially.

---

### Official Review · Reviewer_dibr · 2021-10-30

**Correctness:** 3
**Technical Novelty And Significance:** 2
**Empirical Novelty And Significance:** 3
**Recommendation:** 8
**Confidence:** 3

**Main Review:**

**Update:** After reading the rebuttal and other reviews, I have increased my original score. Thank you to the authors for addressing the concerns raised in great detail.

**Strengths:**
- The problem tackled by the authors is, in my opinion, an exceedingly difficult one.
- The experimental section is extensive and the authors have done their best to analyze different facets of the problem.
- The paper is very carefully written.

**Weaknesses:**
- The usefulness of the method heavily relies on how good the DAG construction is. In this paper, the authors construct a kNN graph to define the DAG, which seems a reasonable choice, but then how sensitive is the overall method to this choice?
- Without confidence bands, many of the results presented are rather speculative. For example, based on the results reported in Table 5, the authors speculate that distal peak-gene interactions may be characterized by longer lags between peaks and genes. While this conclusion seems reasonable, it is rather hard to gauge whether there is a singificant decrease in proximal peak-gene pairs when going from two to ten GNN layers.
- The authors make a decent attempt at putting their work into context, but if they want to focus on the causal aspect, there is a whole literature on causal Bayesian network inference and structural causal models that seems to be missing in the reference section. Apart from that, some other papers that seem relevant:
   - https://www.jmlr.org/papers/volume16/basu15a/basu15a.pdf
   - https://doi.org/10.1093/bioinformatics/btq377
   - https://www.pnas.org/content/115/9/2252.short

Other comments:
- page 8, Table 3: How do you explain the huge increase in performance for GrID-Net (eQTL & Sparse Peaks + Chromatin interactions & Sparse Genes + Chromatin Interactions & Sparse Peaks) and VAR Granger (Chromatin interactions & Sparse Peaks) over the correlation methods for the SHARE-seq data set? Based on Tables 1 and 2 it seems to be related to a sudden increase in AUPRC? Could you comment on that?
- page 11, title of reference by Heinz et al. is incorrect
- page 13, DAG construction: I would be careful when normalizing the data when used for causal inference because the normalization might destroy those relationships. Here I understand that the DAG is determined post-normalization, so that should be okay, since these relationships are inferred.

**Summary Of The Paper:**

The authors propose a method for modeling single-cell multimodal data using Granger causality. They introduce a graph neural network called GrID-Net (Granger Inference on DAGs), where they assume that the network variables (the cells) follow a DAG structure. They focus on applying their method to the task of inferring non-coding genomic loci that causally influence the expression of a specific gene.

**Summary Of The Review:**

Overall the method is sound and the authors have put in a lot of work to showcase the merits of their approach. I wouldn't say that the proposed approach "outperforms existing methods for inferring regulatory locus-gene links", but it seems to be an incremental step forward. Keeping in mind that the problem tackled is an exceedingly difficult one, I am inclined to give this paper the benefit of a doubt.

---

> ### Author Response · Authors · 2021-11-23
> **Detailed Responses to Reviewer #3 (dibr)**
>
> We thank you for your feedback and below are detailed responses to your comments. Please also see a summary of the major revisions above.
>
> **Comment:** The usefulness of the method heavily relies on how good the DAG construction is. In this paper, the authors construct a kNN graph to define the DAG, which seems a reasonable choice, but then how sensitive is the overall method to this choice?
>
> **Response:** We thank the reviewer for recognizing the importance of the information encoded by the DAG in the GrID-Net framework. We assessed the robustness of GrID-Net with respect to the two components that underlie the DAG construction: the initial kNN graph construction and the orientation of edges with respect to pseudotime. First, we modified the number of nearest neighbors used in generating the kNN graph and found GrID-Net’s performance on the eQTL and chromatin interaction prediction tasks to be robust to this parameter. Second, we assessed the performance of GrID-Net when using different state-of-the-art pseudotime inference methods, including diffusion pseudotime, Monocle and Palantir. We found that GrID-Net was also robust to these different choices of state-of-the-art pseudotime inference methods (“DAG construction” in main text; Appendix A.2 “DAG Robustness Analysis”).
>
> **Comment:** Without confidence bands, many of the results presented are rather speculative. For example, based on the results reported in Table 5, the authors speculate that distal peak-gene interactions may be characterized by longer lags between peaks and genes. While this conclusion seems reasonable, it is rather hard to gauge whether there is a singificant decrease in proximal peak-gene pairs when going from two to ten GNN layers.
>
> **Response:** We agree with the reviewer on the value of including confidence bands in presenting our results. We have now updated all of our results to include the mean and standard deviation of values associated with GrID-Net and GVAR, both of which are neural network-based approaches for which results can be affected by different parameter initializations. The other benchmarks are completely deterministic and their performance remains constant across runs.
>
> **Comment:** The authors make a decent attempt at putting their work into context, but if they want to focus on the causal aspect, there is a whole literature on causal Bayesian network inference and structural causal models that seems to be missing in the reference section.
>
> **Response:** We appreciate the reviewer’s insight into related works that will help contextualize our present work and have added additional references to these important works in the “Related Work” section.
>
> **Comment:** How do you explain the huge increase in performance for GrID-Net (eQTL & Sparse Peaks + Chromatin interactions & Sparse Genes + Chromatin Interactions & Sparse Peaks) and VAR Granger (Chromatin interactions & Sparse Peaks) over the correlation methods for the SHARE-seq data set? Based on Tables 1 and 2 it seems to be related to a sudden increase in AUPRC? Could you comment on that?
>
> **Response:** We appreciate the reviewer’s consideration of the unique aspects of the various datasets that contribute to these results. For example, sci-CAR was one of the first RNA+ATAC multimodal studies, and is more noisy and sparse than SHARE-seq, which was more recently introduced (Ma et al., 2020). Also, direct comparisons of a method’s performance on the full candidate peak-gene set vs. the sparse set are complicated by varying true positive rates between the full set and the sparse subset. As such we believe it is more appropriate to compare relative performance across methods within each setting, rather than absolute AUPRC scores across settings.
>
> **Comment:** title of reference by Heinz et al. is incorrect
>
> **Response:** We thank the reviewer for identifying this mistake, and we have corrected the reference to this publication.
>
> **Comment:** I would be careful when normalizing the data when used for causal inference because the normalization might destroy those relationships. Here I understand that the DAG is determined post-normalization, so that should be okay, since these relationships are inferred.
>
> **Response:** We are grateful to the reviewer for drawing our attention to the possible effects of various normalization procedures on causal inference. In evaluating GrID-Net against the baseline methods, we sought to use the same preprocessed data for all methods in order to have apples-to-apples comparisons. Accordingly, we applied a uniform set of standard single-cell normalization procedures to the data across all tested methods.

---

> > ### Comment · Reviewer_dibr · 2021-11-29
> > **Thank you for the reply!**
> >
> > Thank you for addressing most of my concerns and those of my fellow reviewers. Regarding the use of relative versus absolute performance measures, I agree that the former might be more useful, although it still seems to me that the performance improvement on SHARE-seq is larger than for the other two data sets. I understand however that there are many aspects to consider and it might be hard to gauge why that happens exactly.
> >
> > After reading the rebuttal and the other reviews, I am slightly more positive about the paper, so I have bumped my score up relative to my original evaluation.

---

> > > ### Author Response · Authors · 2021-11-29
> > > **Thanks for updating the score!**
> > >
> > > We'll continue to investigate the performance variation across datasets and will be sure to update the paper as we gain additional insights.

---

> ### Author Response · Authors · 2021-11-28
> **Thank you for updating the score!**
>
> Your feedback was extremely valuable and helped improve the paper substantially.

---

### Official Review · Reviewer_QYmT · 2021-11-02

**Correctness:** 4
**Technical Novelty And Significance:** 2
**Empirical Novelty And Significance:** 2
**Recommendation:** 8
**Confidence:** 3

**Main Review:**

Detailed Comments:

1. There are some issues with the model or claims about it that we note:
1.a.
There are many instances in the manuscript where the relation between genetic loci and expression is referred to as “casual”. It needs to be made clear that this is “Granger causal” or “temporally causal” as this does not represent true causality. An obvious example would be a true causal factor that affects ATAC peak at time X and later, after some unrelated process takes effect, expression of a gene at time X+t. Clearly the definitions used here would conclude a casual relation that does not exist.
1.b.
Using a t-test to check for differences in means of the full and reduced model seems problematic. A reduced model will always have higher MSE than a full model on the sample data. The metric needs to be adjusted for the number of free parameters/model complexity. An F test would be more appropriate. Alternatively, the authors could evaluate performance on a held out test set. As it stands, this issue makes us doubt the reported results.
1.c.
There is a somewhat misuse of terminology with regards to the L variable defined in the manuscript as this does not represent the same lag as that is defined by Granger causality. The manuscript interprets L correctly but overloads the use of the term “lag”. It would be helpful to show a scenario (i.e reduction) where GrID-Net is equivalent to Granger causality inference using a linear model. For example, using a linear graph (Fig. 1a left) as input with L=1.
1.d.
It was somewhat unclear to us if model implementation and target function in Eq 10 means the parameters of the reduced and full are shared. It seems to be the case. If so, isn’t it possible that the reduced model suffers from the full model optimization problem? I.e. if the reduced model was optimized on its own without shared params it would converge to a better predictive solution?

2. There are several issues with the analysis that we note:
We understand that a technical contribution is of prime importance for ICLR. However, the following suggestions for improving the data analysis could significantly improve the practical utility of the manuscript.
2.a. Data
2.a.i
Using liftover to map human eQTLs to mouse genome is problematic.
2.a.ii
The direction of the eQTL is not clear since they are chosen purely based on a p-value cutoff. Analyzing the relation between variant creation, change in expression and change in ATAC peaks would help clarify what the model is learning (enhancer/repressor creation/disruption).

2.b Performance and Evaluation
2.b.i
The choice of baselines is not optimal. The Var Granger approach will perform poorly since it is just a simplification of GrID-Net. The correlation methods could have been applied to the graph directly to make the comparison more fair given that the graph itself is not part of the proposed method and is created using existing software.
2.b.ii
The model is almost making random predictions (AUROC hovering around 0.5). It also does not improve significantly compared to the baselines. We understand that the relation to eQTL based calls are problematic but with such poor correspondence it brings doubt to using this as a metric.
2.b.iii
The sparsity analysis is ancillary and does not contribute much to the manuscript. First, it is strange that the performance is marginally better on the sparse data (correlation of 0s inflating the result?). Second, without explicitly modeling single cell sparsity, mentioning robustness to sparsity is not productive.

2.c. Mechanistic Analysis
2.c.i
How is a TF associated with a gene? You need to show that the KO of a TF is actually associated with a gene expression increase/decrease. Presence of a TF motif does not always mean the expression is altered.
2.c.ii
There is no enrichment analysis of TF binding. Does the GrID-Net detect more peak-gene pairs overall compared to the baselines? If so, a raw number of detected pairs doesn’t tell us anything about enrichment. A simple fisher test would suffice.
2.c.iii
The interpretation behind the proximal vs distal analysis is not supported biologically. This is an extremely bold claim without experimental evidence. The authors should rephrase this section to be more speculative.
2.c.iv
Consider using enrichment tests here instead of a percentage. It is not clear how different the shown percentage is from the null. For example, is the ratio of distal to proximal labels close to the observed percentage after changing L? If so, the percentage increase is too small to have any biological meaning.

2.d. Lack of clarity
2.d.i
How are the p-value cutoff chosen for classification in AUPRC? Even if a standard 0.05 is used, it should still be mentioned.
2.d.ii
How is Pearson correlation being computed with missing values in the sparsity analysis? If using 0s, this would inflate the correlation.
2.d.iii
How does the model handle multiple terminal lineages (i.e. multiple nodes without any outgoing edges). What is y in this case?

2.e Relation to previous work:
2.e.i
The authors frame the entire work as a novel generalization of Granger causality. While we understand the need to frame the work in terms of ML novelty in a ML conference, we found this to misrepresent the actual work. In practice, Granger causality is not defined exactly as it is here, and the “generalization of Granger causality for DAGs” boils down to standard GNN operations over graphs with a memory of length L, using two competing architectures. The solution seems fine for the problem at hand (given a good answer to the possible issues raised above), but the remote relation to Granger seems to justify not much more than a general comment in the discussion.


UPDATE
==============
We read all of the authors detailed answers and reviewed the updated manuscript. The authors did a very good job addressing many of the concerns and comments raised by us and the other reviewers. While some concerns may remain we acknowledge the good work and effort made by the authors to improve the analysis, clarify details and make more precise claims. We have updated our score accordingly.


**Summary Of The Paper:**

In this work, the authors propose a model framed as an extension to Granger causality for DAGs, and apply their model to infer “causal” relations between changes in gene expression and chromatin accessibility over time. Rather than relying on linear models and ordered data, the authors use a graph neural network to account for partially ordered data (i.e. branching structure). Similar to Granger inference, they compare a full model using paired data to a reduced model which excludes the lagging predictor. The authors then focus on analyzing temporally ordered scRNASeq and scATACSeq in 3 data sets from human and mouse. They test the method on predicting “causality” between the data modalities by using eQTLs and and chromatin interactions as proxy labels and compare against several baseline methods, noting that the method is robust to sparsity in data. Finally, they find enrichment of TF binding motifs in peak-gene pair regions and conclude that the model detects a lag between distal ATACSeq peaks and gene expression.


**Summary Of The Review:**

Overall, the paper introduces a model that, while a simple GNN, which appears to be a very reasonable approach for solving the problem of detecting the effects of chromatin accessibility on expression. The problem itself is interesting and important. In addition, the manuscript is generally clear and well written though the authors should be more careful about how claims are interpreted and described. However, the work does not offer many novel technical innovations since the GNN architecture is standard and the domain specific modeling/insights is lacking. Also, aspects of the evaluation of the model are concerning since the authors do not use out of sample data and should consider other baseline.

---

> ### Author Response · Authors · 2021-11-23
> **Detailed Responses to Reviewer #2 (QYmT)**
>
> We thank you for your feedback and below are detailed responses to your comments. Please also see a summary of the major revisions above.
>
> **Comment:** There are many instances in the manuscript where the relation between genetic loci and expression is referred to as “casual”. It needs to be made clear that this is “Granger causal” or “temporally causal” as this does not represent true causality. An obvious example would be a true causal factor that affects ATAC peak at time X and later, after some unrelated process takes effect, expression of a gene at time X+t. Clearly the definitions used here would conclude a casual relation that does not exist.
>
> **Response:** We fully agree that our approach does not infer true causality and have sought to make that clear. We have now edited the manuscript and changed all potentially ambiguous references to “causality” as “Granger” or “temporal causality”.  We also agree that our approach cannot account for hidden confounders and have added a discussion about it in the “Related Work” section. As we discuss in this section, by identifying a shortlist of high-confidence Granger causal peak-gene relationships, GrID-Net enables downstream biological validation via perturbational studies.
>
> **Comment:** An F test would be more appropriate. Alternatively, the authors could evaluate performance on a held out test set. As it stands, this issue makes us doubt the reported results.
>
> **Response:** We thank the reviewer for this insightful critique and agree that F-tests are more standard in Granger causality. Accordingly, we have changed our method to use F-tests instead of the t-test. The results in the main text have been correspondingly updated. We found that these results did not change materially. In fact, the use of the F-test actually led to a very slight improvement overall (Section 2.3 in main text; comparison of F-test and t-test in Appendix A.7). As we describe in A.7, we believe the two tests yield similar results because the test statistic is used only to rank candidate peak-gene pairs. In that regard, the two tests yield very similar rankings. We note that since each peak-gene pair is evaluated under the same setup, the free parameters and model complexity are held constant across them and only the residual errors change.
>
> We also appreciate the reviewer’s suggestion of using a held out test set, but unfortunately this approach is not suitable, given that the dynamics represented in a dataset’s DAG would be disrupted by holding out cells in a manner similar to eliminating observations from a time series in traditional Granger causal analyses. As such, we consider the use of the F test to be a superior option.
>
>
> **Comment:** There is a somewhat misuse of terminology with regards to the L variable defined in the manuscript as this does not represent the same lag as that is defined by Granger causality. The manuscript interprets L correctly but overloads the use of the term “lag”. It would be helpful to show a scenario (i.e reduction) where GrID-Net is equivalent to Granger causality inference using a linear model. For example, using a linear graph (Fig. 1a left) as input with L=1.
>
> **Response:** We thank the reviewer for pointing out this confusion. We have now edited the manuscript to clarify that L does not represent “lag” as used in Granger causality. We have also added clarification in Section 2 to show how a linear graph of Fig 1a would be implemented in GrID-Net.
>
> **Comment:** It was somewhat unclear to us if model implementation and target function in Eq 10 means the parameters of the reduced and full are shared. It seems to be the case. If so, isn’t it possible that the reduced model suffers from the full model optimization problem? I.e. if the reduced model was optimized on its own without shared params it would converge to a better predictive solution?
>
> **Response:** We apologize for the confusion. The parameters of the full and reduced model are completely independent as we have clarified. When performing the joint optimization, the total loss is the sum of losses of the full and reduced models. Hence, the optimization process, in seeking to minimize the sum, will minimize each term separately. Just to confirm that this was indeed the case, we also ran some test cases of optimizing the full and reduced models separately and achieved near-identical results (Appendix A.8)

---

> > ### Author Response · Authors · 2021-11-23
> > **Detailed Responses to Reviewer #2 (QYmT) - Part 2**
> >
> > **Comment:** Using liftover to map human eQTLs to mouse genome is problematic
> >
> > **Response:** We respectfully disagree, and believe this is a reasonable approach in the face of limited eQTL data availability in mouse. As we describe in the paper (“Predicting eQTLs”; also Appendix A.5), the eQTL data needs to be tissue-specific but, unfortunately, limited mouse eQTL data is available. On the other hand,  the epigenome is largely conserved across mouse and human, providing theoretical support for our data-transfer approach (see citations in A.5). RNA+ATAC-seq multimodal studies are still in their infancy and as more datasets become available for human tissue, we expect the need for such a data-transfer approach will be reduced.
> >
> > **Comment:** Direction of eQTL influence and of peak-gene connection
> >
> > **Response:** We appreciate your insightful observation and believe this could be a powerful extension of the approach proposed in this work. Here, we have limited ourselves to requiring a change in expression, without focusing on the direction of change. Given the amount and sparsity of the data available, we do not believe there currently is sufficient data to confidently ascertain direction on top of peak-gene links. However, as the quantity and quality of data improves, future work could seek to additionally identify activation/repression relationships for each pair and evaluate them against the direction of eQTLs.
> >
> > **Comment:** 2.b.i The choice of baselines
> >
> > **Response:** We strove to incorporate baselines based on literature (the correlation based approaches) while also introducing strong Granger causality baselines. Regarding the removal of zeros, we followed previous work in computing correlation estimates and kept the zeros during computation. We note that the “Pseudocell Correlation” approach does take into account the kNN graph and averages observations across a group of cells, so it should have more robustness than the per-cell correlation measure. We apologize if this was unclear and have edited the text.
> >
> > To address the concern that GrID-Net might be acquiring its performance solely by being nonlinear, we have added a new benchmark named GVAR, a state-of-the-art neural network-based nonlinear Granger causal inference technique. As we show, GrID-Net outperforms GVAR as well, indicating that the DAG-based approach adds value.
> >
> > **Comment:** 2.b.ii The model is almost making random predictions (AUROC hovering around 0.5).
> >
> > **Response:** We note that the performance of GrID-Net, as well as that of the baseline approaches, varies across the different datasets. We believe this is due to differences in data quality and robustness. sci-CAR data is lower quality, as it was one of the first demonstrations of RNA+ATAC multimodal readouts, while the more recently introduced SHARE-seq method has higher quality data. Indeed, on SHARE-seq data, GrID-Net achieves substantially stronger performance.
> >
> > That said, we fully agree that there remains substantial room for improvement on this problem. As Reviewer #3 mentioned, the biological problem we address here is “an exceedingly difficult one.” In that regard, we believe we present a notable performance improvement upon the state-of-the-art, while also introducing an innovative approach to the problem.

---

> > > ### Author Response · Authors · 2021-11-23
> > > **Detailed Responses to Reviewer #2 (QYmT) - Part 3**
> > >
> > > **Comment:** 2.b.iii The sparsity analysis is ancillary and does not contribute much to the manuscript. First, it is strange that the performance is marginally better on the sparse data (correlation of 0s inflating the result?). Second, without explicitly modeling single cell sparsity, mentioning robustness to sparsity is not productive.
> > >
> > > **Response:** Regarding the second point, we believe that a discussion of sparsity during the evaluation is valuable, given the inherent sparsity and noisiness of measurements from current single-cell multimodal assays. Our approach effectively averages over cells in a DAG-defined neighborhood, helping it to achieve robustness. We have now included in the “Results” section the intuition underlying how GrID-Net’s design contributes to its robust performance in sparse settings. We respectfully disagree that an explicit model of single-cell sparsity is always required-- there is substantial precedent in single-cell research (e.g., “metacell” approaches) for such neighborhood-averaging based approaches to *implicitly* address sparsity and noise.
> > >
> > > With regards to the performance of the various methods in the sparse settings, we note that the proportions of true positive eQTLs or chromatin-interactions over the entire peak-gene candidate set (Tables 1 and 2) differ from those on its respective sparse-data subset (Table 3). As such, we have included a table with baseline proportions of true positives in the Appendix (A.9) to contextualize these differences. In particular, we believe that the higher AUPRC numbers for GrID-Net (and also for some of the baselines) are a function of differing eQTL and chromatin-interaction data availability for peak-gene pairs that are represented sparsely in single-cell multimodal data. Accordingly, it is more meaningful to compare relative performance within each table, rather than performance across tables.
> > >
> > > **Comment:** 2.c.i How is a TF associated with a gene? You need to show that the KO of a TF is actually associated with a gene expression increase/decrease. Presence of a TF motif does not always mean the expression is altered.
> > >
> > > **Response:** For each gene in the top ranking peak-gene pairs, we associate it with putative TFs by searching for TF motifs in the associated peaks. We agree that biological validation (e.g,. via a knock-out screen) would be the gold-standard to confirm the existence of a TF-gene regulatory relationship. However, we believe that new biological validation is out of the scope of an ICLR paper. Instead, we take a computational approach: noting that TF binding motifs have been widely used as proxies for the potential functional relevance of noncoding regulatory regions, we applied the tool Homer to identify the TFs that had multiple motifs in the peaks associated with a gene. Our use of Homer implements the intuition that, for a putative TF-target pair, a high number of enriched motifs of the TF located in noncoding regions linked to the target gene is a strong indication of their putative regulatory interaction. We apologize if this was unclear and have edited the text to clarify it further.
> > >
> > > To benchmark GrID-Net on this measure, we perform an apples-to-apples comparison between it and the baselines. For each method, we selected the top 1% of candidate peak-gene pairs and applied the procedure above. Our results indicated that, across these peak-gene sets of the same size, GrID-Net’s results correspond to a significantly greater number of TF motifs.
> > >
> > > Furthermore, we have newly added a set of analyses in the “Results” section that further demonstrates the enhanced functional relevance of peak-gene pairs detected by GrID-Net by comparing peaks prioritized in the various methods with TF binding sites from ChIP-seq data.
> > >
> > > **Comment:** 2.c.ii Enrichment analysis of TF binding.
> > >
> > > **Response:** We thank the reviewer for pointing out a potential area of confusion in this part of the analysis and have clarified in the caption for Table 4 that the number of peak-gene pairs used for this analysis is identical across the various methods. We also appreciate the reviewer’s suggestion to assess the statistical significance of differences in the detected motifs associated with the various methods. We evaluated the significance of GrID-Net’s results against each of the alternative methods, corroborate GrID-Net’s significant outperformance (p < 10^-10 in all cases, one-sided binomial test with Bonferroni correction) . We have reported the use of this test and the statistical significance of these comparisons in the “Results” section.

---

> > > > ### Author Response · Authors · 2021-11-23
> > > > **Detailed Responses to Reviewer #2 (QYmT) - Part 4**
> > > >
> > > > **Comment:** 2.c.iii and 2.c.iv  Proximal vs distal analysis
> > > >
> > > > **Response:** We thank the reviewer for pointing out that this section needed to be clarified. Recognizing the speculative nature of this exploration (though we note GrID-Net newly enables such a study), we have moved much of this discussion to the Appendix A.11, with just a brief discussion in the main text.
> > > >
> > > > In Appendix A.11, we have added more results. In particular, across the various choices of $L$, the same number of top hits were chosen (top 1% of candidate pairs in each dataset). Thus, the increase in distal peak-gene pairs with higher $L$ is not merely due to more hits being reported.
> > > >
> > > > **Comment:** 2.d.i How are the p-value cutoff chosen for classification in AUPRC?
> > > >
> > > > **Response:** We are sorry that this was not clear. We do not use a p-value cutoff; instead, we use the test-statistic to rank all candidate pairs. The precision-recall and ROC curves are computed in the standard way by varying the threshold for choosing the top hits and computing the precision, recall, sensitivity and specificity at each threshold. We directly used the Python library scikit-learn’s functions for this computation. We have now updated the main text (“Methods”) to clarify this.
> > > >
> > > > **Comment:** 2.d.ii How is Pearson correlation being computed with missing values in the sparsity analysis?
> > > >
> > > > **Response:** The simpler Pearson correlation baseline is computed using the same approach that’s used in the existing literature, which involves estimating the correlation on normalized counts data and considers missing values as zeros. The pseudocell correlation baseline also follows previously-described approaches in the literature: it is computed by averaging observations for cells in a kNN-graph neighborhood, thus somewhat mitigating the issue of missing values.
> > > >
> > > > **Comment:** 2.d.iii How does the model handle multiple terminal lineages (i.e. multiple nodes without any outgoing edges). What is y in this case?
> > > >
> > > > **Response:** In the GrID-Net framework, gene expression measurements (y) for terminal nodes in the DAG are predicted by the measurements of the ancestors of these terminal nodes. In applying GrID-Net to datasets that include multiple terminal lineages, the inferred peak-gene pairs describe a global set of interactions that encompass the full dataset. We note that GrID-Net offers users the flexibility of inferring peak-gene pairs that are specific to a particular lineage by applying GrID-Net only to the subgraph of the DAG that corresponds to that lineage.
> > > >
> > > > **Comment:** 2.e Relation to previous work: 2.e.i The authors frame the entire work as a novel generalization of Granger causality. While we understand the need to frame the work in terms of ML novelty in a ML conference, we found this to misrepresent the actual work. In practice, Granger causality is not defined exactly as it is here, and the “generalization of Granger causality for DAGs” boils down to standard GNN operations over graphs with a memory of length L, using two competing architectures. The solution seems fine for the problem at hand (given a good answer to the possible issues raised above), but the remote relation to Granger seems to justify not much more than a general comment in the discussion.
> > > >
> > > > **Response:** We respectfully disagree. We have now updated our approach to use F-tests, as is more standard in Granger causal analyses, and have also compared with a state-of-the-art nonlinear Granger causality approach, finding that the outperformance of GrID-Net persists. We believe that GrID-Net’s relative strength comes from combining the pseudotime considerations of VAR Granger and GVAR approaches, with the kNN-graph based cell neighborhoods (see note in “Benchmarking”). There has been innovative work toward incorporating neural networks for Granger causal inference and our work is a contribution towards that.

---

> > > > > ### Comment · Reviewer_QYmT · 2021-11-29
> > > > > **Response to Authors answers and updated manuscript**
> > > > >
> > > > > We read all of the authors detailed answers and reviewed the updated manuscript. The authors did a very good job addressing many of the concerns and comments raised by us and the other reviewers. While some concerns may remain (e.g. scalability raised by Reviewer #1) this is to be expected and should not deter from the overall good work and effort made by the authors to improve the analysis, clarify details and make more precise claims. We are happy to update our score accordingly.

---

> > > > > > ### Author Response · Authors · 2021-11-29
> > > > > > **Thank you for updating the score!**
> > > > > >
> > > > > > Your feedback was extremely valuable and helped improve the paper substantially.

---

### Official Review · Reviewer_xWaf · 2021-11-03

**Correctness:** 4
**Technical Novelty And Significance:** 3
**Empirical Novelty And Significance:** Not applicable
**Recommendation:** 8
**Confidence:** 4

**Main Review:**

Strengths
•	Integrative and generalized modeling of DNN and granger causal inference over DAG

•	The authors provided details and reasonable justifications on three datasets, data processing, training & hyperparameters and pseudotiming analysis for building DAGs

•	The benchmark is comprehensive and convincing, providing different metrics to evaluate GrID vs. other state-of-arts

•	Validation of gene-region interactions by independent eQTLs and Hi-C data

Weaknesses
•	The authors probably need to elaborate on the GNN architecture to represent historical information, ideally via visualization on Fig 1b. Also, do Hy and Hx use the same architecture (e.g., # of layers)?

•	Is it possible to compare DAGs by alignment-based methods such as manifold alignment methods? For instance, UnionCom, MMD-MA, etc use manifolds to align scRNA-seq & scATAC-seq and uncover pseudotiming trajectories. I suggest that the authors also benchmark pseudotiming approaches.

•	How cell-type specific are those causal gene-region interactions? Can they better classify cell types than gene expression/chromatin accessibility only? Also, the authors still used the bulk tissue eQTLs data in GTEx that can be confounded by different cell types.

•	The authors should discuss scalability, esp. on single-cell data applications.


**Summary Of The Paper:**

This paper developed an exciting method, GrID-net to infer granger causality between multi-modality on directed acyclic graph (DAG) in the framework of deep neural network (GNN), aiming to improve identifying causal interactions across modalities. Primarily, the authors first find the DAG among samples and represent the historical information of modal features via aggregating GNN layers over DAG. They then apply granger causal inference to test the causal interactions of any two features if adding such GNN-based historical information of one feature significantly helps reduce the prediction loss of another feature.

Also, it is nice that the authors applied GrID-net to multiple recent single-cell multi-modal datasets (scRNA-seq and scATAC-seq) and identified casual chromatin regions for gene expression at the cell type level (grander causal gene-region interactions), providing novel gene regulatory mechanistic insights. Besides, the authors validated gene-region interactions by eQTLs, Hi-C, and TFBS enrichment.


**Summary Of The Review:**

In general, the manuscript was well written and organized logically. The GrID-net modeling was also presented clearly. I believe the new model has broad and timely impact on single cell data integration and understanding temproal gene regulatory mechanisms.

---

> ### Author Response · Authors · 2021-11-23
> **Detailed Responses to Reviewer #1 (xWaf)**
>
> We thank you for your feedback and below are detailed responses to your comments. Please also see a summary of the major revisions above.
>
> **Comment:** The authors probably need to elaborate on the GNN architecture to represent historical information, ideally via visualization on Fig 1b. Also, do Hy and Hx use the same architecture (e.g., # of layers)?
>
> **Response:** We thank the reviewer and agree that adding more visualization details to Figure 1 substantially improves it. The submodels corresponding to Hx(full), Hy(full), and Hy(reduced) use identical architectures and we have now clarified this in the text (“Training Details and Hyperparameters”).
>
>
> **Comment:** Is it possible to compare DAGs by alignment-based methods such as manifold alignment methods? For instance, UnionCom, MMD-MA, etc use manifolds to align scRNA-seq & scATAC-seq and uncover pseudotiming trajectories. I suggest that the authors also benchmark pseudotiming approaches.
>
> **Response:** We appreciate the reviewer’s thoughtful consideration of various single-cell approaches that can be used with GrID-Net. As recommended by them, we evaluated different state-of-the-art pseudotime inference algorithms (i.e. diffusion pseudotime, Palantir, Monocle) for constructing DAGs, finding that GrID-Net is robust to the choice of pseudotime method. It is also robust to the choice of *k* in the kNN graph construction. We have addressed this in the main text (“DAG Construction”) and added further details in the Appendix (“DAG Robustness Analysis”).
>
> We agree that manifold alignment approaches (like UnionCom by Cao et al., now cited in “DAG Construction”) could be a valuable enhancement. These methods could be especially useful when additional features beyond gene expression and chromatin accessibility are available and need to be incorporated. However, we believe their fruitful integration into GrID-Net poses a challenge that is best addressed in future work: GrID-Net is designed to relate the RNA-seq data of a cell to ATAC-seq data of other cells earlier in pseudotime, a task that is currently most suitable for assays that can profile both features in the same cell. Manifold alignment approaches would need to be adapted to be cognizant of this temporal lag and not overly emphasize same-cell alignment of RNA-seq and ATAC-seq.
>
> **Comment:** How cell-type specific are those causal gene-region interactions? Can they better classify cell types than gene expression/chromatin accessibility only? Also, the authors still used the bulk tissue eQTLs data in GTEx that can be confounded by different cell types.
>
> **Response:** We appreciate the reviewer’s interest in the cell type-specificity of both the peak-gene pairs inferred from GrID-Net as well as the eQTL data. As GrID-Net operates over a DAG that captures a population of cells across various cell type and/or states, the peak-gene pairs that are inferred using GrID-Net are specific to the unique cell types/states that are represented in the DAG. We have included a description of the cell type-specificity of the inferred peak-gene pairs in the “Datasets and Preprocessing” subsection in the main text and also provided recommendations for adapting datasets for use with GrID-Net so as to infer peak-gene pairs specific to particular cell types of interest.
>
> With regards to the cell type-specificity of the eQTL data, we sought to use eQTLs derived from tissues that most closely resembled the specific cell types that represented each single-cell multimodal dataset, as eQTL data for the exact cell type(s)/cell lines captured by each dataset is not currently available.
>
> **Comment:** The authors should discuss scalability, esp. on single-cell data applications.
>
> **Response:** We apologize that this was unclear in the submission. We have now added further details on handling large single-cell datasets in the “Runtime and Memory Usage” section of the Appendix, complementing our original discussion on the general scalability of GrID-Net. The size of adjacency matrices in the GNN layers scales as O(n^2) where n is the number of cells. This poses the key scalability challenge, both in terms of GPU memory as well as run-time. Accordingly, we have included a suggestion that sketching or sampling approaches, which we cite in the paper, be used to preprocess input data before applying GrID-Net. Here, manifold alignment could also be applied to identify a shared feature space over which to compute the sketching.

---

> ### Author Response · Authors · 2021-11-29
> **Thank you for updating the score!**
>
> Your review was very helpful.

---

### Author Response · Authors · 2021-11-23
**To All Reviewers: Summary of Revisions**

We thank all the reviewers for their detailed feedback and careful reading of the manuscript. We have substantially revised the paper to address each of their concerns and believe it has greatly improved as a result. We summarize the major revisions here and provide detailed responses to each of the reviews separately below:

**Major revisions:**
- New evaluations of our method against ChIP-seq data; these evaluations agree with findings from previous analyses.
- Use of the F-test instead of the t-test for full-vs.-reduced model selection, as recommended by reviewer #2.
- More comprehensive citations of related work on causal analysis, and a discussion of our approach’s limitations, as recommended by reviewers #3 and #4
- Additional baseline for benchmarking: a state-of-the-art, nonlinear Granger approach, as recommended by reviewers #2 and #4.
- Additional robustness analysis re data preprocessing, pseudotime computation etc., as recommended by reviewers #1 and #3
- Confidence bands of AUPRC/AUROC scores across multiple runs of evaluation, as recommended by reviewers #3 and #4.

---

> ### Author Response · Authors · 2021-11-28
> **List of all revisions to the manuscript**
>
> [update on 11/28]: For the convenience of the reviewers, we're listing below all the revisions:
>
> **Introduction**
>  -  Fig 1b: additional visualization to clarify the method (reviewer #1)
>  -  replace references to "causal" inference with "Granger causal"/"temporally causal" inference (reviewer #2)
>
> **Methods**
>  -  clarify "lag" in a few instances where it was unclear/inappropriate (reviewer #2)
>  -  for clarity, describe when GriD-Net inference would reduce to standard Granger time-series inference i.e. when the DAG is a linear graph (reviewer #2)
>  - [Major revision]: redo *all* results in the paper by using the F-test, instead of the t-test, for model selection (reviewer #2). In Appendix A.7, we show that results from the two model selection approaches were very similar and discuss why.
>  -  clarify the separation of parameters between full and reduced models (reviewers #1 and #2)
>  -  experiments to show that simultaneously minimizing the sum of losses of full and reduced models yields similar results as minimizing them separately (reviewer #2)
>  -  clarify that architectures H_x, H_y, H_z are identical but separate (reviewer #1)
>  -  discuss the scalability constraints of the algorithm and ways to address it by sampling/sketching (reviewer #1)
>
> **Related Work**
>  -  [Major revision]: broader coverage of Granger causal literature (incl. citations suggested by reviewers #3 and #4)
>
> **Results**
>  - discussion of cell types in the multimodal datasets chosen, and cell-type specific nature of GrID-Net predictions (reviewer #1). Relatedly, choice of bulk eQTL data specific to each tissue
>  - discuss the possible use of manifold alignment algorithms in computing the kNN graph (reviewer #1)
>  - [Major revision] experiments to evaluate alternative pseudotime algorithms as well as different choices of k in kNN graph (reviewers #1 and #3)
>  - [Major revision] addition of the new non-linear Granger causal benchmark (GVAR) (reviewers #2 and #4)
>  - clarifications on the other baselines, e.g., robustness of the Pseudocell approach (reviewer #2)
>  - contextualize our method's performance gains on sparse data, along with an intuitive justification (reviewers #2 and #3)
>  - [Major revision] addition of mean and standard deviations for all evaluations (reviewers #3 and #4)
>  - justification for use of liftOver in acquiring eQTL data (reviewer #2)
>  - clarification on how transcription factor (TF) motifs were associated with peaks (reviewer #2)
>  - [Major revision] new experiments to validate predictions against cell type-specific ChIP-seq data, supporting the findings from TF binding motif experiments.
>  - addition of statistical tests of significance for TF binding motif results (reviewer #2)
>  - additional supporting results (and confidence bands) for the analysis on relating genomic distance to temporal lag, i.e., differences in discovery rates for distal vs. proximal peaks (reviewers #2 and #3). However, this analysis is less pertinent to our method's evaluation than the new ChiP-seq results. Accordingly, in order to make room for the new ChIP-seq results, we compressed this sub-section's text and moved the tables etc. to the appendix.

---

### Author Response · Authors · 2021-11-23
**To the AC, SAC and all reviewers**

We are grateful to the reviewers for their thoughtful comments, and have strived hard to address them all by new experiments, better discussion of related work, and clearer writing. We believe the draft is much improved as a result and would appreciate the reviewers' consideration of our responses and manuscript changes.

With regards to some of the reviewers' comments below about our lack of response, we submitted our responses prior to the rebuttal submission deadline of Nov 22, 11:59pm AoE, as described in a recent email by the ICLR PCs. We apologize if there was any misunderstanding on the timing of this deadline.

---

### Decision · Program_Chairs · 2022-01-20

**Decision:**

Accept (Poster)

**Comment:**

The AC and reviewers all agree that the paper proposes a very interesting framework to extend Granger Causality to DAG structured dynamical systems with important applications.

The submission was the object of extensive discussion, and the AC and reviewers all agree that the author feedback satisfactorily addresses the vast majority of their concerns. We strongly urge the authors to incorporate all the points and revisions mentioned in their feedback.

We certainly hope that the author will pursue this line of work and consider scaling their approach to tackle larger applications such as those related to social networks.